# eXponential FAmily Dynamical Systems (XFADS): Large-scale nonlinear Gaussian state-space modeling

**Matthew Dowling**
Champalimaud Research, Champalimaud Foundation, Portugal
matthew.dowling@research.fchampalimaud.org

**Yuan Zhao**
National Institute of Mental Health, USA
yuan.zhao@nih.gov

**Il Memming Park**
Champalimaud Research, Champalimaud Foundation, Portugal
memming.park@research.fchampalimaud.org

## Abstract

State-space graphical models and the variational autoencoder framework provide a principled apparatus for learning dynamical systems from data. State-of-the-art probabilistic approaches are often able to scale to large problems at the cost of flexibility of the variational posterior or expressivity of the dynamics model. However, those consolidations can be detrimental if the ultimate goal is to learn a generative model capable of explaining the spatiotemporal structure of the data and making accurate forecasts. We introduce a low-rank structured variational autoencoding framework for nonlinear Gaussian state-space graphical models capable of capturing dense covariance structures that are important for learning dynamical systems with predictive capabilities. Our inference algorithm exploits the covariance structures that arise naturally from sample based approximate Gaussian message passing and low-rank amortized posterior updates – effectively performing approximate variational smoothing with time complexity scaling linearly in the state dimensionality. In comparisons with other deep state-space model architectures our approach consistently demonstrates the ability to learn a more predictive generative model. Furthermore, when applied to neural physiological recordings, our approach is able to learn a dynamical system capable of forecasting population spiking and behavioral correlates from a small portion of single trials.

## 1  Introduction

State-space models (SSM) are invaluable for understanding the temporal structure of complex natural phenomena through their underlying dynamics [1–3]. While engineering or physics problems often assume the dynamical laws of the system of interest are known to a high degree of accuracy, in an unsupervised data-driven investigation, they have to be learned from the observed data. The variational autoencoder (VAE) framework makes it possible to jointly learn the parameters of the state-space description and an inference network to amortize posterior computation of the unknown latent state [4–7]. However, it can be challenging to structure the variational approximation and design an inference network that permits fast evaluation of the loss function (evidence lower bound or ELBO) while preserving the temporal structure of the posterior.

38th Conference on Neural Information Processing Systems (NeurIPS 2024).

In this work, we develop a structured variational approximation, approximate ELBO, and inference network architecture for generative models specified by nonlinear dynamical systems with Gaussian state noise. Our main contributions are as follows,

(i) A structured amortized variational approximation that combines the prior dynamics with low-rank data updates to parameterize Gaussian distributions with dense covariance matrices,

(ii) Conceptualizing the approximate smoothing problem as an approximate filtering problem for pseudo-observations that encode a representation of current and future data, and,

(iii) An inference algorithm that scales $\mathcal{O}(TL(Sr + S^2 + r^2))$ – made possible by exploiting the low-rank structure of the amortization network as well as Monte Carlo integration of the latent state through the dynamics.

## 2 Background

State-space models are probabilistic graphical models where observations $\mathbf{y}_t$ in discrete time are conditionally independent given a continuous latent state, $\mathbf{z}_t$, evolving according to Markovian dynamics, so that the complete data likelihood for $T$ consecutive observations factorizes as,

$$p(\mathbf{y}_{1:T}, \mathbf{z}_{1:T}) = p_{\boldsymbol{\theta}}(\mathbf{z}_1)\, p_{\boldsymbol{\psi}}(\mathbf{y}_1 \,|\, \mathbf{z}_1) \times \prod_{t=2}^{T} p_{\boldsymbol{\psi}}(\mathbf{y}_t \,|\, \mathbf{z}_t)\, p_{\boldsymbol{\theta}}(\mathbf{z}_t \,|\, \mathbf{z}_{t-1})$$

where $\mathbf{z}_t \in \mathbb{R}^L$ are real-valued latent states, $\boldsymbol{\theta}$ parameterizes the dynamics and initial condition, and $\boldsymbol{\psi}$ parameterizes the observation model. When the generative model, $(\boldsymbol{\theta}, \boldsymbol{\psi})$, is known, the statistical inference problem is to compute the smoothing posterior, $p(\mathbf{z}_{1:T} \,|\, \mathbf{y}_{1:T})$[2]. Otherwise, $(\boldsymbol{\theta}, \boldsymbol{\psi})$ have to be learned from the data – known as system identification[8].

Variational inference makes it possible to accomplish these goals in a harmonious way. The variational expectation maximization (vEM) algorithm iterates two steps: first, we maximize a lower bound to the log-marginal likelihood – the ELBO – with respect to the parameters of an approximate posterior, $q(\mathbf{z}_{1:T}) \approx p(\mathbf{z}_{1:T} \,|\, \mathbf{y}_{1:T})$; then, with the approximate posterior fixed, the ELBO is maximized with respect to parameters of the generative model[9]. For large scale problems, vEM can be slow due to the need to fully optimize the variational parameters before taking gradient steps on parameters of the generative model. Therefore, the variational autoencoder (VAE) is better suited for large scale problems for its ability to simultaneous learn the generative model and inference network – an expressive parametric function that maps data to the parameters of approximate posterior[10,11].

**Model specifications.** Although our approach is applicable to any exponential family state-space process, given their ubiquity, we focus on dynamical systems driven by Gaussian noise so that,

$$p_{\boldsymbol{\theta}}(\mathbf{z}_t \,|\, \mathbf{z}_{t-1}) = \mathcal{N}(\mathbf{z}_t \,|\, \mathbf{m}_{\boldsymbol{\theta}}(\mathbf{z}_{t-1}), \mathbf{Q}_{\boldsymbol{\theta}}) \tag{1}$$

where $\mathbf{m}_{\boldsymbol{\theta}} : \mathbb{R}^L \to \mathbb{R}^L$ might be a nonlinear neural network function with learnable parameters $\boldsymbol{\theta}$, and $\mathbf{Q}_{\boldsymbol{\theta}} \in \mathbb{R}^{L \times L}$ is a learnable state-noise covariance matrix. Given the favorable properties of exponential family distributions[12-15], especially in the context of variational inference, we write the prior dynamics in their exponential family representation (natural parameter form),

$$p_{\boldsymbol{\theta}}(\mathbf{z}_t \,|\, \mathbf{z}_{t-1}) = h(\mathbf{z}_t) \exp\left(\mathcal{T}(\mathbf{z}_t)^{\top} \boldsymbol{\lambda}_{\boldsymbol{\theta}}(\mathbf{z}_{t-1}) - A(\boldsymbol{\lambda}_{\boldsymbol{\theta}}(\mathbf{z}_{t-1}))\right) \tag{2}$$

where $h$ is the base measure, $\mathcal{T}(\mathbf{z}_t)$ the sufficient statistics, $A(\cdot)$ the log-partition function, and $\boldsymbol{\lambda}_{\boldsymbol{\theta}}(\cdot)$ is a map $\mathbb{R}^L \mapsto \mathbb{R}^{L^2+L}$ that transforms $\mathbf{z}_{t-1}$ to natural parameters for $\mathbf{z}_t$. For a Gaussian distribution, the sufficient statistics can be defined as $\mathcal{T}(\mathbf{z}_t)^{\top} = \begin{bmatrix} \mathbf{z}_t^{\top} & -\frac{1}{2}\mathbf{z}_t\mathbf{z}_t^{\top} \end{bmatrix}$, so that $\boldsymbol{\lambda}_{\boldsymbol{\theta}}(\cdot)$ for (1) is given by,

$$\boldsymbol{\lambda}_{\boldsymbol{\theta}}(\mathbf{z}_{t-1}) = \begin{bmatrix} \mathbf{Q}_{\boldsymbol{\theta}}^{-1}\mathbf{m}_{\boldsymbol{\theta}}(\mathbf{z}_{t-1}) \\ \mathbf{Q}_{\boldsymbol{\theta}}^{-1} \end{bmatrix} \qquad \text{(dynamics model in natural paramter form)} \tag{3}$$

As it will simplify subsequent analysis, the *mean parameter* mapping corresponding to this natural parameter mapping (guaranteed to exist as long as the exponential family is minimal[12]) is given by

$$\boldsymbol{\mu}_{\boldsymbol{\theta}}(\mathbf{z}_{t-1}) = \mathbb{E}_{p_{\boldsymbol{\theta}}(\mathbf{z}_t|\mathbf{z}_{t-1})}\left[\mathcal{T}(\mathbf{z}_t)\right] = \begin{bmatrix} \mathbf{m}_{\boldsymbol{\theta}}(\mathbf{z}_{t-1}) \\ -\frac{1}{2}\left(\mathbf{m}_{\boldsymbol{\theta}}(\mathbf{z}_{t-1})\mathbf{m}_{\boldsymbol{\theta}}(\mathbf{z}_{t-1})^{\top} + \mathbf{Q}_{\boldsymbol{\theta}}\right) \end{bmatrix} \qquad \begin{pmatrix} \text{mean} \\ \text{parameter} \\ \text{form} \end{pmatrix} \tag{4}$$

Furthermore, we make the following assumptions **i)** the state-noise, $\mathbf{Q}_{\boldsymbol{\theta}}$, is diagonal or structured for efficient matrix-vector multiplications. **ii)** $\mathbf{m}_{\boldsymbol{\theta}}(\cdot)$, is a nonlinear smooth function. **ii)** the likelihood, $p_{\boldsymbol{\psi}}(\mathbf{y}_t \,|\, \mathbf{z}_t)$, may be non-conjugate. **iv)** $L$ may be large enough so that $L^3$ is comparable to $T$.

**Amortized inference for state-space models.** A useful property of SSMs is that, $\mathbf{z}_t$ conditioned on $\mathbf{z}_{t-1}$ and $\mathbf{y}_{t:T}$, is independent of $\mathbf{y}_{1:t-1}$, i.e., $p(\mathbf{z}_t \,|\, \mathbf{z}_{t-1}, \mathbf{y}_{1:T}) = p(\mathbf{z}_t \,|\, \mathbf{z}_{t-1}, \mathbf{y}_{t:T})$[5,16]. It thus suffices to construct an approximate posterior that factorizes forward in time,

$$q(\mathbf{z}_{1:T}) = q(\mathbf{z}_1) \prod q(\mathbf{z}_t \,|\, \mathbf{z}_{t-1}) \tag{5}$$

and introduce learnable function approximators to amortize inference by mapping $\mathbf{z}_{t-1}$ and $\mathbf{y}_{t:T}$ to the parameters of $q(\mathbf{z}_t \,|\, \mathbf{z}_{t-1})$. This makes it simple to sample $\mathbf{z}_{1:T}$ from the approximate posterior (using the reparameterization trick) and evaluate the ELBO (a.k.a. negative variational free energy),

$$\mathcal{L}(q) = \sum \mathbb{E}_{q_t} \left[ \log p(\mathbf{y}_t \,|\, \mathbf{z}_t) \right] - \mathbb{E}_{q_{t-1}} \left[ \mathbb{D}_{\mathrm{KL}}(q(\mathbf{z}_t \,|\, \mathbf{z}_{t-1}) || \, p_{\boldsymbol\theta}(\mathbf{z}_t \,|\, \mathbf{z}_{t-1})) \right] \leq \log p(\mathbf{y}_{1:T}) \tag{6}$$

where $\mathbb{D}_{\mathrm{KL}}(\cdot || \cdot)$ is the Kullback-Leibler (KL) divergence and $\mathbb{E}_{q_t} \equiv \mathbb{E}_{\mathbf{z}_t \sim q(\mathbf{z}_t; \mathbf{y}_{1:T})}$, so that the generative model and inference network parameters can be learned through stochastic backpropagation. Many works for Gaussian $q(\mathbf{z}_t \,|\, \mathbf{z}_{t-1})$, such as Krishnan et al.[6], Alaa and van der Schaar[17], Girin et al.[18], Hafner et al.[19], construct inference networks that parameterize the variational posterior as

$$q(\mathbf{z}_t \,|\, \mathbf{z}_{t-1}) = \mathcal{N}(\mathbf{m}_{\boldsymbol\phi}(\mathbf{z}_{t-1}, \mathbf{y}_{1:T}), \mathbf{P}_{\boldsymbol\phi}(\mathbf{z}_{t-1}, \mathbf{y}_{1:T})). \tag{7}$$

There are limitless ways to construct $\mathbf{m}_{\boldsymbol\phi}(\cdot)$ and $\mathbf{P}_{\boldsymbol\phi}(\cdot)$ so $\boldsymbol\phi$ can be learned through gradient ascent on the ELBO, but a straightforward and illustrative approach[6,17] is to transform future data, $\mathbf{y}_{t:T}$, using a recurrent neural network (RNN), or any efficient autoregressive sequence to sequence model, and then mapping the preceding latent state, $\mathbf{z}_{t-1}$, using a feed-forward neural network, so that a complete inference network description could be,

$$\left( \mathbf{m}_{\boldsymbol\phi}(\mathbf{z}_{t-1}, \mathbf{y}_{t:T}), \mathbf{P}_{\boldsymbol\phi}(\mathbf{z}_{t-1}, \mathbf{y}_{t:T}) \right) = \mathrm{NN}([\mathbf{z}_{t-1}, \mathbf{u}_t]), \qquad \mathbf{u}_t = \mathrm{S2S}([\mathbf{u}_{t+1}, \mathbf{y}_t]) \tag{8}$$

where $\mathrm{S2S}(\cdot)$ is a parametric sequence-to-sequence function that maintains a hidden state $\mathbf{u}_t$ and takes as input $\mathbf{y}_t$, and $\mathrm{NN}(\cdot)$ is a parametric function designed to output approximate posterior parameters. This leads to a backward-forward algorithm, meaning that data $\mathbf{y}_{1:T}$ are mapped to $\mathbf{u}_{1:T}$ in reverse time, and then samples are drawn from $q(\mathbf{z}_t \,|\, \mathbf{z}_{t-1})$ forward in time.

Possible drawbacks of this inference framework are **i)** missing observations obstruct inference (the example networks cannot naturally accommodate missing data); **ii)** sampling entire trajectories to approximate the expected KL term can potentially lead to high-variance gradient estimators, and **iii)** statistics of the marginals (e.g. second moments) can only be approximated through sample averages.

## 3 Related works

Many existing works also explore inference and data-driven learning for state-space graphical models within the VAE framework. We highlight the most closely related studies and note specific limitations that our work seeks to address. The structured variational autoencoder (SVAE)[20] makes it possible to efficiently evaluate the ELBO while preserving the temporal structure of the posterior by restricting the prior to a *linear dynamical system* (LDS) and then constructing the approximation as $q(\mathbf{z}_{1:T}) \propto p_{\boldsymbol\theta}(\mathbf{z}_{1:T}) \prod \exp(t(\mathbf{z}_t)^\top \psi(\mathbf{y}_t))$ so that its statistics can be obtained using efficient message passing algorithms. However, the SVAE is not directly applicable when the dynamics are nonlinear since the joint prior will no longer be Gaussian (thereby not allowing for efficient conjugate updates). Recently, Zhao and Linderman[21] expanded on the SVAE framework by exploiting the LDS structure and associative scan operations to improve its scalability.

The deep Kalman filter (DKF)[6] uses black-box inference networks to make drawing joint samples from the full posterior simple. However, pure black-box amortization networks such as those can make learning the parameters of the generative model dynamics difficult because their gradients will not propagate through the expected log-likelihood term[5]. In contrast, we consider inference networks inspired by the fundamental importance of the prior for evaluating Bayesian conjugate updates. The deep variational Bayes filter (DVBF) also considers inference and learning in state-space graphical models[5]. Difficulties of learning the generative model that arise as a result of more standard VAE implementations defining inference networks independent of the prior are handled by forcing samples from the approximate posterior to traverse through the dynamics. Our work extends this concept, by directly specifying the parameters of the variational approximation in terms of the prior.

Our approach constructs an inference network infused with the prior similar to the SVAE and DVBF but i) avoids restrictions to LDS and ii) affords easy access to approximations of the marginal statistics (such as the dense latent state covariances) without having to average over sampled trajectories (or store them directly which would be prohibitive as the latent dimensionality becomes large).

# 4  Method

An alternative to constructing variational approximations through specification of conditional distributions, as in Eq. (7), involves the use of data-dependent Gaussian potentials, that we refer to as pseudo-observations:

$$p(\tilde{\mathbf{y}}_t \,|\, \mathbf{z}_t) \propto \exp(\tilde{\boldsymbol{\lambda}}_\phi(\mathbf{y}_{1:T})^\top \mathcal{T}(\mathbf{z}_t)) \equiv \exp(\tilde{\boldsymbol{\lambda}}_t^\top \mathcal{T}(\mathbf{z}_t)) = \exp\left(\mathbf{k}_t^\top \mathbf{z}_t - \tfrac{1}{2}||\mathbf{K}_t \mathbf{z}_t||^2\right) \tag{9}$$

These Gaussian potentials can then be combined with the prior through Bayes' rule, yielding the approximation

$$q(\mathbf{z}_{1:T}) = \frac{\prod p(\tilde{\mathbf{y}}_t \,|\, \mathbf{z}_t) p_{\boldsymbol{\theta}}(\mathbf{z}_{1:T})}{p(\tilde{\mathbf{y}}_{1:T})} \tag{10}$$

A benefit of this formulation, is that it inherently imposes the latent dependency structure of the generative model onto the amortized posterior. This parameterization, was introduced in Johnson et al.[20], where an important point highlighted, is that the Gaussian potentials can encode any arbitrary subset of observations; for example, $p(\tilde{\mathbf{y}}_t \,|\, \mathbf{z}_t)$ could be made to depend on $\mathbf{y}_t$ alone, or even the entire dataset, $\mathbf{y}_{1:T}$. Regardless of that particular design choice, the corresponding ELBO for the variational approximation of Eq. (10) is

$$\mathcal{L}(q) = \sum \mathbb{E}_{q_t}\left[\log p(\mathbf{y}_t \,|\, \mathbf{z}_t)\right] - \mathbb{E}_{q_t}\left[\log p(\tilde{\mathbf{y}}_t \,|\, \mathbf{z}_t)\right] + \log p(\tilde{\mathbf{y}}_{1:T}) \tag{11}$$

For linear Gaussian latent dynamics, conjugate potentials could be efficiently integrated with the prior using exact message passing, yielding filtered and smoothed marginal statistics. The smoothed statistics can be used to evaluate the first two terms on the right-hand side, while the filtered statistics can be used to evaluate the final term, the log-marginal likelihood of the pseudo-observations.

However, this approach does not directly apply to nonlinear dynamical systems, where the variational posterior is no longer Gaussian. Since evaluating the smoothed marginals and the log-marginal likelihood of pseudo-observations relies on first obtaining the filtered marginals, a logical starting point is to develop a method for approximating these filtered marginals. To this end, we propose a differentiable approximate message passing algorithm specifically designed to compute filtered posterior statistics in models characterized by nonlinear latent dynamics and observations represented as Gaussian potentials. Building on this foundation, we then return to the subsequent challenges of efficiently computing smoothed posterior statistics and evaluating the ELBO.

**Differentiable nonlinear filtering.** Bayesian filtering is often conceptualized as a two step procedure[2]. In the *predict* step, our belief of the latent state is integrated through the dynamics, yielding $q(\mathbf{z}_t \,|\, \tilde{\mathbf{y}}_{1:t-1}) = \int p_{\boldsymbol{\theta}}(\mathbf{z}_t \,|\, \mathbf{z}_{t-1}) q(\mathbf{z}_{t-1} \,|\, \tilde{\mathbf{y}}_{1:t-1}) \, \mathrm{d}\mathbf{z}_{t-1}$ (a.k.a. the predictive distribution). Then, applying Bayes' rule, $q(\mathbf{z}_t \,|\, \tilde{\mathbf{y}}_{1:t}) \propto p(\tilde{\mathbf{y}}_t \,|\, \mathbf{z}_t) q(\mathbf{z}_t \,|\, \tilde{\mathbf{y}}_{1:t-1})$, we *update* our belief. Evidently then, developing an approximate filtering algorithm that exploits the conjugacy of the pseudo observations can be recast as the problem: given an approximation $\pi(\mathbf{z}_{t-1}) \approx q(\mathbf{z}_{t-1} \,|\, \tilde{\mathbf{y}}_{1:t-1})$, find an approximation to the predictive distribution, $\bar{\pi}(\mathbf{z}_t) \approx q(\mathbf{z}_t \,|\, \tilde{\mathbf{y}}_{1:t-1})$. The recursion would continue forward by updating our belief analytically, setting $\pi(\mathbf{z}_t) \propto p(\tilde{\mathbf{y}}_t \,|\, \mathbf{z}_t)\bar{\pi}(\mathbf{z}_t)$, then finding a Gaussian approximation of $\bar{\pi}(\mathbf{z}_{t+1})$ and so forth.

With the problem restated this way, we propose an approximate filtering solution designed to exploit two key factors at play **i)** the approximate beliefs are constrained to the same exponential family as the latent state transitions **ii)** the pseudo observations are encoded as conjugate potentials. Our approach involves recursively solving intermediary variational problems (their fixed point solutions on the right),

Variational filtering

(i) $\bar{\pi}(\mathbf{z}_t) = \arg\min \, \mathbb{D}_{\mathrm{KL}}\big(\mathbb{E}_{\pi_{t-1}}\left[p_{\boldsymbol{\theta}}(\mathbf{z}_t \,|\, \mathbf{z}_{t-1})\right]\big|\big| \, \bar{\pi}(\mathbf{z}_t)\big) \implies \bar{\boldsymbol{\mu}}_t = \mathbb{E}_{\pi_{t-1}}\left[\boldsymbol{\mu}_{\boldsymbol{\theta}}(\mathbf{z}_{t-1})\right]$  (12)

(ii) $\pi(\mathbf{z}_t) = \arg\min \, \mathbb{D}_{\mathrm{KL}}(\pi(\mathbf{z}_t) || \, p(\tilde{\mathbf{y}}_t \,|\, \mathbf{z}_t)\bar{\pi}(\mathbf{z}_t)) \implies \boldsymbol{\lambda}_t = \bar{\boldsymbol{\lambda}}_t + \tilde{\boldsymbol{\lambda}}_t$  (13)

Steps (i) and (ii) can be thought of as variational analogues of the predict/update steps of Bayesian filtering, and importantly, finding their fixed point solutions does not require an iterative procedure because of our problem specifications. Reassuringly, iterating (i) and (ii) in the case of an LDS generative model would exactly recover the information form Kalman filtering equations. In the case of nonlinear dynamical systems, directly taking the expectation in (i) is intractable. We can overcome

this by employing the reparameterization trick to obtain a differentiable sample approximation of the parameters, $\bar{\boldsymbol{\mu}}_t$, of the fixed point solution. Naturally now, the statistics of the approximate filtered beliefs can be used to approximate the log-marginal likelihood of the pseudo observations as,

$$\log p(\tilde{\mathbf{y}}_{1:T}) = \sum \log \int p(\tilde{\mathbf{y}}_t \,|\, \mathbf{z}_t) q(\mathbf{z}_t \,|\, \tilde{\mathbf{y}}_{1:t-1}) \, \mathrm{d}\mathbf{z}_t \approx \sum \log \int p(\tilde{\mathbf{y}}_t \,|\, \mathbf{z}_t) \bar{\pi}(\mathbf{z}_t) \, \mathrm{d}\mathbf{z}_t \qquad (14)$$

where the last integral can be evaluated analytically as a result of the Gaussian forms of the approximations and pseudo observations. While formulating step (ii) as a variational problem may appear superfluous given the conjugate structure, it hints at the possibility of approximating smoothed posterior marginal statistics within a variational framework. However, pursuing this idea further reveals significant challenges. Trying to develop a backward recursion for the distribution $q_t$ minimizing $\mathbb{D}_{\mathrm{KL}}\big(\mathbb{E}_{q_{t+1}}[q_{t|t+1}]\,\big\|\, q_t\big)$, by using the forward KL divergence (as in step (i)), leads to an intractable problem because the backward Markov transitions, $q_{t|t+1}$, are not conditionally Gaussian. Conversely, a fixed point solution of the reverse KL objective (as in step(ii)), $\mathbb{D}_{\mathrm{KL}}\big(q_t\,\big\|\, \mathbb{E}_{q_{t+1}}[q_{t|t+1}]\big)$, necessitates an iterative procedure, which can be computationally expensive.

**Smoothing as filtering.** In light of these difficulties, we offer a simple solution that exploits the flexibility in choosing the pseudo observation data dependence: define the parameters, $\mathbf{k}_t$ and $\mathbf{K}_t$, of each pseudo observation, $\tilde{\mathbf{y}}_t$, to be a function of current *and* future data, $\mathbf{y}_{t:T}$, so that,

$$p(\tilde{\mathbf{y}}_t \,|\, \mathbf{z}_t) \propto \exp(\tilde{\boldsymbol{\lambda}}_\phi(\mathbf{y}_{t:T})^\top \mathcal{T}(\mathbf{z}_t)) \qquad (15)$$

With this choice, filtered statistics of the latent state—relative to the pseudo-observations—can be used to approximate posterior smoothed marginals, i.e. $\pi(\mathbf{z}_t) \approx q(\mathbf{z}_t \,|\, \tilde{\mathbf{y}}_{1:t}) \approx p(\mathbf{z}_t \,|\, \mathbf{y}_{1:T})$. This solution circumvents the challenges associated with backward message computation and only requires a single pass through the pseudo observations to obtain approximate smoothed posterior statistics. Substituting, $\pi_t$, as an approximation to $q_t$, in Eq. (11), leads to the following approximation of the ELBO.

---

— Variational smoothing ELBO —

$$\hat{\mathcal{L}}(\pi) = \sum \mathbb{E}_{\pi_t} \left[\log p(\mathbf{y}_t \,|\, \mathbf{z}_t)\right] - \mathbb{E}_{\pi_t} \left[\log p(\tilde{\mathbf{y}}_t \,|\, \mathbf{z}_t)\right] + \log \mathbb{E}_{\bar{\pi}_t} \left[p(\tilde{\mathbf{y}}_t \,|\, \mathbf{z}_t)\right] \qquad (16)$$

$$= \sum \mathbb{E}_{\pi_t} \left[\log p(\mathbf{y}_t \,|\, \mathbf{z}_t)\right] - \mathbb{D}_{\mathrm{KL}}(\pi(\mathbf{z}_t)\,\|\, \bar{\pi}(\mathbf{z}_t)) \qquad (17)$$

---

By expressing the approximate ELBO compactly as Eq.(17), we highlight that it promotes learning models where the posterior at time $t$ aligns closely with the one-step posterior predictive at that time (which depends on the generative model and the posterior at time $t-1$). The KL term in Eq.(17) can be evaluated in closed form, while the expected log-likelihood term can be approximated using the reparameterization trick[4]. However, a point of practical importance should be raised now: every filtering step and evaluation of the KL term has a time complexity of $\mathcal{O}(L^3)$, which may become a bottleneck for large $L$. In the following discussion, we will explore effective strategies to parameterize the Gaussian potential inference network that produces $\tilde{\boldsymbol{\lambda}}_{1:T}$, to reduce the computational burden that filtering and evaluating $\hat{\mathcal{L}}(\pi)$ pose in the large $L$ regime.

**Local and backward encoders.** For state-space models, inferences about the latent state should be possible even with missing observations. To enable the amortized inference network to process missing observations in a principled way, we decompose the natural parameter update into two additive components: **i)** a *local* encoder, $\boldsymbol{\alpha}_\phi(\cdot)$, for current observation, and **ii)** a *backward* encoder, $\boldsymbol{\beta}_\phi(\cdot)$, for future observations, i.e.,

$$\tilde{\boldsymbol{\lambda}}_\phi(\mathbf{y}_{t:T}) = \boldsymbol{\alpha}_\phi(\mathbf{y}_t) + \boldsymbol{\beta}_\phi(\mathbf{y}_{t+1:T}) \quad \text{(or for the sake of brevity)} \quad \tilde{\boldsymbol{\lambda}}_t = \boldsymbol{\alpha}_t + \boldsymbol{\beta}_{t+1} \qquad (18)$$

Furthermore, by building the dependence of $\boldsymbol{\beta}_\phi(\cdot)$ on $\mathbf{y}_{t+1:T}$ through their representation as $\boldsymbol{\alpha}_{t+1:T}$, so that $\boldsymbol{\beta}_\phi(\mathbf{y}_{t+1:T}) = \boldsymbol{\beta}_\phi(\boldsymbol{\alpha}_{t+1:T})$, a missing observation at time $t$ is handled by setting $\boldsymbol{\alpha}_t = \mathbf{0}$. While a data dependent natural parameter update of $\mathbf{0}$ faithfully represents a missing observation – in the absence of data, the prior should not be updated – alternatively setting $\mathbf{y}_t = \mathbf{0}$ would introduce a harmful inductive bias into the inference network, since an observation of $\mathbf{0}$ can be arbitrarily informative. Given the impracticality of $\mathcal{O}(TL^2)$ memory requirements, it is appealing to consider a low-rank parameterization for the local and backward encoders – we consider

$$\boldsymbol{\alpha}_t = \begin{pmatrix} \mathbf{a}_t \\ \mathbf{A}_t \mathbf{A}_t^\top \end{pmatrix} := \begin{pmatrix} \mathbf{a}(\mathbf{y}_t) \\ \mathbf{A}(\mathbf{y}_t)\mathbf{A}(\mathbf{y}_t)^\top \end{pmatrix} \qquad \boldsymbol{\beta}_t = \begin{pmatrix} \mathbf{b}_t \\ \mathbf{B}_t \mathbf{B}_t^\top \end{pmatrix} := \begin{pmatrix} \mathbf{b}(\boldsymbol{\alpha}_{t:T}) \\ \mathbf{B}(\boldsymbol{\alpha}_{t:T})\mathbf{B}(\boldsymbol{\alpha}_{t:T})^\top \end{pmatrix} \qquad (19)$$

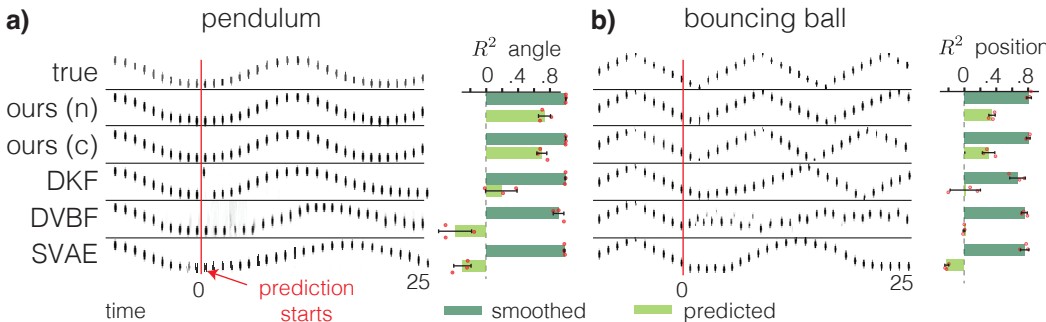

Figure 1: **Smoothing and predictive performance on bouncing ball and pendulum.** To the left of the red line are samples from the posterior during the data window projected to image space, to the right of the red line are samples unrolled from $p_{\boldsymbol{\theta}}(\mathbf{z}_t \mid \mathbf{z}_{t-1})$. **a)** while all methods are adept at smoothing in the context window, our methods predictive performance is better by a noticeable margin as measured by the $R^2$. **b)** similar results hold for the bouncing ball dataset.

where $\mathbf{A}_t \in \mathbb{R}^{L \times r_\alpha}$ with $\mathbf{B}_t \in \mathbb{R}^{L \times r_\beta}$ parameterize low-rank local/backward precision updates, and $\mathbf{a}_t \in \mathbb{R}^L$ with $\mathbf{b}_t \in \mathbb{R}^L$ parameterize local/backward precision-scaled mean updates. Using these descriptions and the additive decomposition (18), the parameters of a single pseudo observation are,

$$\tilde{\boldsymbol{\lambda}}_t = \begin{pmatrix} \mathbf{k}_t \\ \mathbf{K}_t \mathbf{K}_t^\top \end{pmatrix} := \begin{pmatrix} \mathbf{k}(\mathbf{y}_{t:T}) \\ \mathbf{K}(\mathbf{y}_{t:T}) \mathbf{K}(\mathbf{y}_{t:T})^\top \end{pmatrix} = \begin{pmatrix} \mathbf{a}_t + \mathbf{b}_t \\ [\mathbf{A}_t \ \mathbf{B}_t][\mathbf{A}_t \ \mathbf{B}_t]^\top \end{pmatrix} \tag{20}$$

where $\mathbf{K} \in \mathbb{R}^{L \times r}$ if $r = r_\alpha + r_\beta$ and $\mathbf{k} \in \mathbb{R}^L$. The low-rank structure of the natural parameter updates will be a key component to develop an efficient approximate message passing algorithm for obtaining sufficient statistics of the approximate posterior and evaluating the ELBO. Analogous to the inference network description (8), a differentiable architecture producing $\boldsymbol{\alpha}_{1:T}$ and $\boldsymbol{\beta}_{1:T}$ could be,

$$\boldsymbol{\alpha}_t = \text{NN}(\mathbf{y}_t) \qquad\qquad \boldsymbol{\beta}_t = \text{S2S}([\boldsymbol{\beta}_{t+1}, \boldsymbol{\alpha}_t]), \tag{21}$$

which overall defines the map $\mathbf{y}_{1:T} \mapsto (\boldsymbol{\alpha}_{1:T}, \boldsymbol{\beta}_{1:T})$. In addition, the separation of local and backward encoders can reduce the complexity of the backward encoder for $L < N$. Those familiar with sequential Monte-Carlo (SMC) methods[22] can view the backward encoder similar to twisting functions used to combine future information with filtered state beliefs to produce smoothing approximations[23,24].

**Exploiting structure for efficient filtering.** A benefit of using the forward KL to design a variational analogue to the exact Bayesian predict step is immediate access to the fixed point solution. While nonlinear specification of the latent dynamical system make the expectation of Eq. (12) intractable, using the reparameterization trick with $\mathbf{z}_{t-1}^s \sim \pi(\mathbf{z}_{t-1})$, gives the approximation,

$$\bar{\boldsymbol{\mu}}_t = {}^1\!/\!s \sum_{s=1}^S \begin{bmatrix} \mathbf{m}_{\boldsymbol{\theta}}(\mathbf{z}_{t-1}^s) \\ -\frac{1}{2} \left( \mathbf{m}_{\boldsymbol{\theta}}(\mathbf{z}_{t-1}^s) \mathbf{m}_{\boldsymbol{\theta}}(\mathbf{z}_{t-1}^s)^\top + \mathbf{Q}_{\boldsymbol{\theta}} \right) \end{bmatrix} \tag{22}$$

where $S$ is the total number of samples. Converting this finite sample estimate from mean parameter coordinates to a mean/covariance representation, we get that,

$$\bar{\mathbf{m}}_t = {}^1\!/\!s \sum_{s=1}^S \mathbf{m}_{\boldsymbol{\theta}}(\mathbf{z}_{t-1}^s) \qquad\qquad \bar{\mathbf{P}}_t = \bar{\mathbf{M}}_t^c \bar{\mathbf{M}}_t^{c\top} + \mathbf{Q}_{\boldsymbol{\theta}} \tag{23}$$

where $\bar{\mathbf{M}}_t^c$ is the $L \times S$ matrix of samples passed through the dynamics function, then centered by the mean, defined for convenience as,

$$\bar{\mathbf{M}}_t^c = {}^1\!/\!\sqrt{S} \left[ \mathbf{m}_{\boldsymbol{\theta}}(\mathbf{z}_{t-1}^1) - \bar{\mathbf{m}}_t, \cdots, \mathbf{m}_{\boldsymbol{\theta}}(\mathbf{z}_{t-1}^S) - \bar{\mathbf{m}}_t \right] \in \mathbb{R}^{L \times S} \tag{24}$$

Writing the covariance estimate as it is in Eq. (23), reveals that it can alternatively be represented by the pair $(\bar{\mathbf{M}}_t^c, \mathbf{Q}_{\boldsymbol{\theta}})$. In the regime where $L > S$, significant computational savings can be afforded by capitalizing on the low-rank structure of the covariance as estimated via the reparameterization trick. This structure can be exploited for efficient linear algebraic operations involving $\bar{\mathbf{P}}_t$ (and its inverse, after application of the Woodbury identity). Consequently, the natural parameters of $\pi_t$, after updating $\bar{\pi}_t$, are

$$\mathbf{P}_t^{-1} \mathbf{m}_t = \bar{\mathbf{P}}_t^{-1} \bar{\mathbf{m}}_t + \mathbf{k}_t \qquad\qquad \mathbf{P}_t^{-1} = \bar{\mathbf{P}}_t^{-1} + \mathbf{K}_t \mathbf{K}_t^\top \tag{25}$$

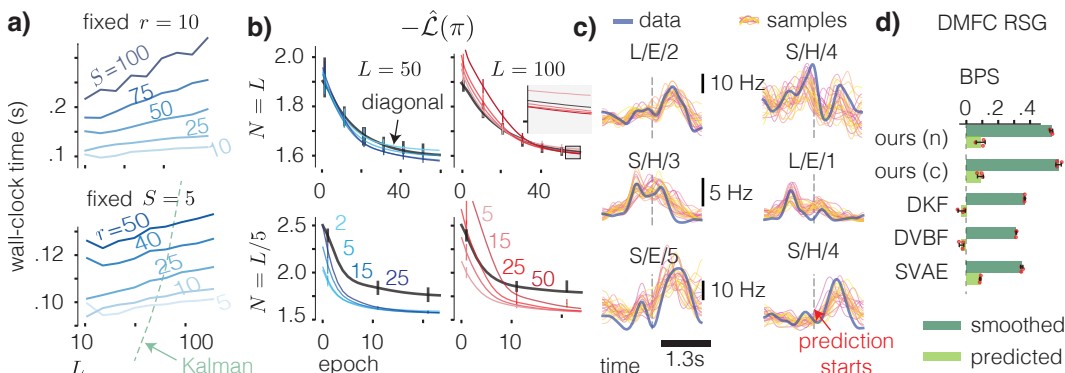

Figure 2: **a)** Empirical time complexity scaling. Since complexity is a function of $L$, $S$, and $r$, we vary $L$ (top) for fixed $r = 10$ and (bottom) for fixed $S = 5$; we examine several values of the variable not fixed. Examining wall-clock time shows empirically our implementation scales linearly in $L$; on the (bottom) we plot wall-clock time for a Kalman filter implementation, showing the standard cubic dependence on $L$. **b)** (top) Negative ELBO as a function of training epoch when $N = L$ (bottom) when $N = L/5$; the left column shows the case $L = 50$ and the right when $L = 100$. Different colors indicate different settings of the local/backward encoder rank; zooming in for $L = 100$, shows low-rank updates can match diagonal ones. **c)** Peristimulus time histogram (PSTH) for the DMFC RSG dataset for different trial condition averages; we consider a context window of 1.3s and a prediction window of 1.3s. **d)** BPS for each method for context/prediction windows.

and also admit structured representations. Without *both* the sample approximation structure (during the variational predict step) and low-rank parameterization (during the variational update step), the cost of approximate filtering each time-step would be dominated by an $\mathcal{O}(L^3)$ cost. Instead, recognizing the potential computational advantages of exploiting these structures, and never instantiating the predictive/updated covariance and precision matrices, makes it possible to develop an approximate filtering algorithm, where in the case $L$ is significantly larger than $S$ or $r$, has complexity of $\mathcal{O}(L(Sr + r^2 + S^2))$ per step. More details regarding time complexity are in App. B.5.

**Efficient sampling and ELBO evaluation.** When $\mathbb{E}_{\pi_t}[\log p(\mathbf{y}_t \,|\, \mathbf{z}_t)]$ can not be evaluated in closed form, Monte-Carlo integration can be used as a differentiable approximation. To sample from $\pi(\mathbf{z}_t)$ without explicitly constructing $\mathbf{P}_t$, we can take $\bar{\mathbf{z}}_t^s \sim \mathcal{N}(\mathbf{0}, \bar{\mathbf{P}}_t)$ and $\mathbf{w}_t^s \sim \mathcal{N}(\mathbf{0}, \mathbf{I}_{L+S})$ and set,

$$\mathbf{z}_t^s = \mathbf{m}_t + \bar{\mathbf{z}}_t^s - \mathbf{K}_t \boldsymbol{\Upsilon}_t \boldsymbol{\Upsilon}_t^\top (\mathbf{K}_t^\top \bar{\mathbf{z}}_t^s + \mathbf{w}_t^s). \tag{26}$$

While more details are provided in App. B.4, this can be done efficiently since samples can be drawn cheaply from $\bar{\pi}(\mathbf{z}_t)$ using Eq. (23). Whereas Monte-Carlo approximations of the expected log-likelihood term might be unavoidable, the closed form solution for the KL between two Gaussian distributions should be used to avoid further stochastic approximations. The only difficulty, is that the time complexity of naively evaluating the KL term,

$$\mathbb{D}_{\mathrm{KL}}(\pi(\mathbf{z}_t) \| \bar{\pi}(\mathbf{z}_t)) = \tfrac{1}{2}\left[(\bar{\mathbf{m}}_t - \mathbf{m}_t)^\top \bar{\mathbf{P}}_t^{-1}(\bar{\mathbf{m}}_t - \mathbf{m}_t) + \mathrm{tr}(\bar{\mathbf{P}}_t^{-1}\mathbf{P}_t) + \log(|\bar{\mathbf{P}}_t|/|\mathbf{P}_t|) - L\right] \tag{27}$$

scales $\mathcal{O}(L^3)$. However, since matrix vector multiplies with $\bar{\mathbf{P}}_t^{-1}$ can be performed efficiently and the trace/log-determinant terms can be rewritten using the square-root factors acquired during the forward pass, as we describe in App. C.1, it is possible to evaluate the KL in $\mathcal{O}(LSr + LS^2 + Lr^2)$ time. After a complete forward pass through the encoded data, we acquire the samples $\mathbf{z}_{1:T}^{1:S}$ and all necessary quantities for efficient ELBO evaluation. We detail the variational filtering algorithm in Alg. 2 in App. C.2 and the complete end-to-end learning procedure in Alg. 1.

**Causal amortized inference for streaming data.** In constructing a fully differentiable variational approximation, the parameters of the approximate marginals were effectively amortized according to a recursion in the natural parameter space by iterating Eqs. (12) and (13). This recursion can be recognized more easily by introducing the function, $\mathfrak{F}_{\boldsymbol{\theta}}(\cdot)$, and writing

$$\boldsymbol{\lambda}_t = \mathfrak{F}_{\boldsymbol{\theta}}(\boldsymbol{\lambda}_{t-1}) + \boldsymbol{\alpha}_t + \boldsymbol{\beta}_{t+1} \quad \text{with} \quad \mathfrak{F}_{\boldsymbol{\theta}}(\boldsymbol{\lambda}_{t-1}) = \nabla A^*\left(\int \pi(\mathbf{z}_{t-1}; \boldsymbol{\lambda}_{t-1}) \boldsymbol{\mu}_{\boldsymbol{\theta}}(\mathbf{z}_{t-1}) \,\mathrm{d}\mathbf{z}_{t-1}\right) \tag{28}$$

---
**Algorithm 1** End-to-end learning
---

> **Input:** $\mathbf{y}_{1:T}$
> **while** not converged **do**
>     **for** $t = T$ **to** $1$ **do**
>         $\boldsymbol{\alpha}_t = \mathrm{NN}(\mathbf{y}_t)$  # local encoder
>         $\boldsymbol{\beta}_t = \mathrm{S2S}([\boldsymbol{\beta}_{t+1}\ \boldsymbol{\alpha}_t])$  # backward encoder
>         $\mathbf{k}_t = \mathbf{a}_t + \mathbf{b}_t$
>         $\mathbf{K}_t = [\mathbf{A}_t\ \mathbf{B}_t]$
>     **end for**
>     $\mathbf{z}_{1:T}^{1:S}, \mathbf{m}_{1:T}, \bar{\mathbf{m}}_{1:T}, \boldsymbol{\Upsilon}_{1:T} = \mathrm{Alg.\ 2}(\mathbf{k}_{1:T}, \mathbf{K}_{1:T})$
>     $\hat{\mathcal{L}}(\pi) = \sum [S^{-1} \sum \log p(\mathbf{y}_t \,|\, \mathbf{z}_t^s) - \mathbb{D}_{\mathrm{KL}}(\pi_t || \bar{\pi}_t)]$
>     $(\boldsymbol{\phi}, \boldsymbol{\theta}, \boldsymbol{\psi}) \leftarrow (\boldsymbol{\phi}, \boldsymbol{\theta}, \boldsymbol{\psi}) - \nabla \hat{\mathcal{L}}(\pi)$
> **end while**
> **Output:** $\mathbf{z}_{1:T}^{1:S}, \mathbf{m}_{1:T}, \bar{\mathbf{m}}_{1:T}, \boldsymbol{\Upsilon}_{1:T}$

---

$\nabla A^* : (\mathbf{m}, -\frac{1}{2}(\mathbf{P} + \mathbf{mm}^\top)) \mapsto (\mathbf{P}^{-1}\mathbf{m}, \mathbf{P}^{-1})$, and $A^*(\cdot)$ is the convex conjugate of the log-partition function[12]. So that, $\mathfrak{F}_{\boldsymbol{\theta}}(\cdot)$ can be thought of as mapping $\boldsymbol{\lambda}_{t-1}$ forward in time by first taking the expectation of (4) with respect to $\pi(\mathbf{z}_{t-1}; \boldsymbol{\lambda}_{t-1})$, and then applying the mean-to-natural coordinate transformation.

One limitation of amortizing inference through the recursion (28) is its inability to produce approximations for the filtering distributions, $p(\mathbf{z}_t \,|\, \mathbf{y}_{1:t})$, which can be valuable in streaming or online settings, as well as for testing hypotheses of causality. However, since (17) only depends on the posterior and posterior predictive marginal statistics, we have the freedom to alter our inference network in a way such that filtered marginal statistics are a by-product of obtaining smoothed marginal statistics. For example, an alternative sequence-to-sequence map for $\boldsymbol{\lambda}_t$ could be defined by,

$$\boldsymbol{\lambda}_t = \mathfrak{F}_{\boldsymbol{\theta}}(\underbrace{\boldsymbol{\lambda}_{t-1} - \boldsymbol{\beta}_t}_{\mathrm{smoothed-future\,=\,filtered}}) + \boldsymbol{\alpha}_t + \boldsymbol{\beta}_{t+1}, \tag{29}$$

so that $\breve{\boldsymbol{\lambda}}_t \equiv \boldsymbol{\lambda}_t - \boldsymbol{\beta}_{t+1}$ obey the recursion $\breve{\boldsymbol{\lambda}}_t = \mathfrak{F}_{\boldsymbol{\theta}}(\breve{\boldsymbol{\lambda}}_{t-1}) + \boldsymbol{\alpha}_t$ and are natural parameters of an approximate filtering distribution, $\breve{\pi}(\mathbf{z}_t) \approx p(\mathbf{z}_t \,|\, \mathbf{y}_{1:t})$. Consequently the approximations to posterior and predictive distributions will have a more complicated relationship than they previously did; while efficient sampling and ELBO evaluation are more intricate as a result – linear time scaling in the state-dimension can still be achieved with additional algebraic manipulations, as we show in App. C.2.

## 5 Experiments

**Time complexity & low-rank precision updates.** We first investigated the properties of low-rank variational Gaussian approximations in the large $L$ regime. To guide us, we had several questions in mind such as: i) how does the performance of low-rank approximations compare to full-rank and diagonal covariance approximations, ii) how large compared to $L$ should the rank of precision updates be to achieve satisfactory results, and iii) how does convergence using low-rank approximations compare to diagonal approximations, considering that they require a larger number of parameters. We expect that full-rank approximations would perform best (given a sufficient amount of data), since the true posterior will have dense second-order statistics due to the interactions of latent states in both the dynamics and observation models. However, it remains unclear how many dimensions are necessary for a low-rank approximation to achieve similar performance and whether this number will be practical.

We simulated data from 50D and 100D linear dynamical systems and compared the convergence between dense and diagonal approximations (Fig. 2**b**); we examined the ELBO for different rank parameterizations in two regimes i) observations and states are of the same dimensionality, $N = L$ (Fig. 2**b** - top), and ii) observations are lower dimensional than states, $N = L/5$ (Fig. 2**b** - bottom). While not surprising that dense variational Gaussian approximations achieve superior performance, message passing in latent Gaussian models with dense covariance scales like $\mathcal{O}(L^3)$[25] and becomes prohibitive for large $L$; thus it is reassuring that in both regimes, low-rank approximate posterior parameterizations achieve comparable results for precision matrix updates of relatively low rank

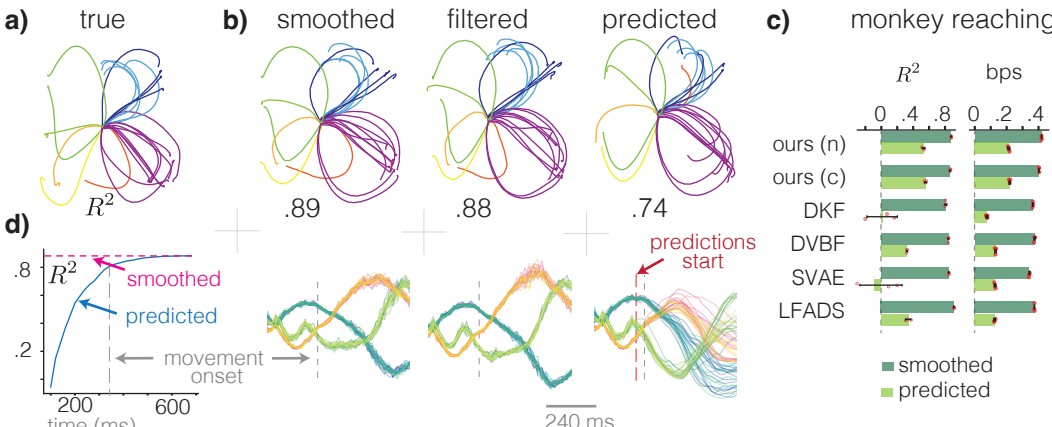

Figure 3: **Predict behavior from a causally inferred initial condition. a)** Actual reaches. **b)** (top) Reaches linearly decoded from smoothed ($R^2 = 0.89$), causally filtered ($R^2 = 0.88$), & predicted ($R^2 = 0.74$) latent trajectories starting from an initial condition causally inferred during the preparatory period. (bottom) Top 3 principal latent dimensions per regime (smoothing/filtering/prediction) for three example trials. **c)** bps / $R^2$ of predicted hand velocity using rates inferred from the 700ms context window and the 500ms prediction window. **d)** Velocity decoding $R^2$ using predicted trajectories as a function of how far into the trial the latent state was filtered until it was only sampled from the autonomous dynamics; by the the movement onset, behavioral predictions using latent trajectory predictions are nearly on par with behavior decoded from the smoothed posterior.

compared to $L$. To empirically examine time complexity scaling as a function of $L$, $S$, and $r$, in Fig. 2**a**, we plot wall-clock times for fixed $r$ while varying $S$ and $L$ (bottom), and for fixed $S$ while varying $r$ and $L$ (top); reassuringly, inference time complexity scales $\mathcal{O}(L)$.

**Baseline comparisons – pendulum & bouncing ball.** Next, we wondered how our approach fared against other modern deep state-space models when it came to learning complex dynamical systems from data. To explore this, we considered two popular datasets: **i)** a pendulum system[26] and **ii)** a bouncing ball[27,28]. Each dataset consists of sequences of observations that are $16 \times 16$ pixel images that are governed by a low-dimensional dynamical system. An interesting aspect of these datasets is that images can be reconstructed with impartial knowledge of the latent state, but for accurate long-term predictions, the dynamics will need to propagate features of the latent state that are irrelevant to the likelihood (e.g. pendulum angular velocity). For benchmarks, three other deep SSM approaches were included: **i)** deep variational Bayes filter(DVBF)[5] **ii)** deep Kalman filter(DKF)[29] **iii)** structured VAE(SVAE)[20]. We denote our causal amortization network with (c) and the noncausal version with (n).

We trained all models in context windows of 50 consecutive images and then sampled future 50 / 25 time-step latent states from the learned dynamical system for pendulum / bouncing ball. To measure quality of learned latent representation and dynamics, we fit angular velocity / position decoders from training set latent states inferred from pendulum / bouncing ball observations. Then, on held-out test data, we measured the $R^2$ of velocity / position predictions during the context (smoothing) and forecast (prediction) windows. Fig. 1 shows that all methods are able to reconstruct well in context windows, however, when prediction is concerned, where the underlying dynamics would need to be learned well for accurate forecasts, our method consistently performs better.

**Neural population dynamics.** We consider two neuroscientific datasets where previous studies have shown the importance of population dynamics in generating plausible hypothesis about underlying neural computation. In addition to DKF, DVBF, and SVAE, we include the LFADS method[7]. First, we considered recordings from motor cortex of a monkey performing a reaching task[30] and evaluate each methods' ability to forecast neural spiking and behavioral correlates. We measure the performance by bits-per-spike (BPS) using inferred spike-train rates[31] and $R^2$ for decoding hand velocity. Similar to the previous experiment, we evaluate the performance in two regimes: **i)** a 700ms context window and **ii)** a 500ms prediction window following an initial context window of 200ms. Fig. 3**c** shows that, while all the methods excel at smoothing in the context window, our method makes more informative

predictions in terms of $R^2$ and BPS. Next, we examined how well the monkey's behavior could be predicted given only causal estimates of the latent state; we trained a model using the causal amortized inference network, given by Eq. (29), then use learned inference network to infer latent states to predict behavior in three regimes: smoothing, filtering, and prediction. Fig. 3**b** shows that hand velocity can be decoded nearly as well in the filtering regime (without access to future data) as in the smoothing regime. In Fig. 3**d**, we plot how the quality of predictions change as filtered latent states are unrolled through the learned dynamics at different points in the trial; showing that forecasts starting prior to movement onset exhibit strong predictive capability.

Next, we investigated our method's performance with data exhibiting a more intricate trial structure. Specifically, we analyzed physiological recordings from the DMFC region of a monkey engaged in a timing interval reproduction task[32]. During this task, the monkey observes a random interval of time (termed the 'ready'-'set' period) demarcated by two cues, and the goal of the monkey is to reproduce that interval (termed the 'set'-'go' period). We perform a similar procedure as before, but for this experiment we use the period before 'set' as the context window, and use the learned dynamics to make predictions onward; in Fig. 2**d** we show the BPS measured on test data during the context and forecast windows. To further investigate the predictive capabilities, we examined condition averaged PSTH produced by samples from the latent state posterior during the joint context/prediction windows. Using the trained model, we sample spike valued observations for the context/prediction windows and then computed condition averaged PSTHs; the results shown in Fig. 2**c**, show that PSTHs sampled from the model remain true to the data, even during the lengthy prediction window.

## 6  Discussion

We presented a new approximate variational filtering/smoothing algorithm, variational learning objective, and Gaussian inference network parameterization for nonlinear state-space models. Focusing on approximations parameterized by dense covariances forced us to consider strategies that ensured computational feasibility for inference and learning with large $L$. The introduced approximate variational filtering algorithm, while especially useful for the nonlinear dynamics we considered, could also be applied to high-dimensional linear systems where exact computations might be infeasible for large $L$. Although our variational objective loses the property of lower-bounding the original data log-marginal likelihood, experiments showed that our method consistently outperforms approaches using potentially tight lower bounds. Quantifying this gap or considering potential corrections present interesting directions for future work. Furthermore, while the same variational approach is applicable to any exponential family dynamical system, specific distributions will have their own associated challenges, offering avenues for further research.

Given that neural computation is inherently nonlinear, system identification methods capable of modeling nonlinear dynamical systems are essential for advancing neuroscience. General SSMs can perform well on inferring smoothed latent state trajectories *without* learning a good model of the nonlinear dynamics. Our proposed method, XFADS, can not only perform efficient system identification and smoothing but also forecast future state evolution for population recordings—a hallmark of a meaningful nonlinear dynamical model; using a causal inference network, XFADS can be used for real-time monitoring, feedback control, and online optimal experimental design, opening the door for new kinds of basic and clinical neuroscience experiments. Future work will focus on developing network architectures for precision matrix updates that are more parameter efficient when the rank those updates become moderate. Moreover, while Alg. 1 remains applicable in the confines of the generative model constraints considered, in certain scenarios, such as when $S > L$, modifications will need to be made to Alg. 2 to minimize time complexity.

## Acknowledgements

We would like to thank the anonymous reviewers and Scott Linderman for their insightful comments and constructive feedback, which greatly improved the quality of this paper. We thank Mahmoud Elmakki for help with organizing the code base and developing helpful demos. Yuan Zhao was supported in part by the National Institute of Mental Health Intramural Research Program (ZIC-MH002968). This work was supported by NIH RF1-DA056404 and the Portuguese Recovery and Resilience Plan (PPR), through project number 62, Center for Responsible AI, and the Portuguese national funds, through FCT – Fundação para a Ciência e a Tecnologia – in the context of the project UIDB/04443/2020.

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

# Contents

## A  Nomenclature

| Symbol | Description |
|---|---|
| SSM | state-space model |
| LGSSM | linear and Gaussian state-space model |
| $\pi(\mathbf{z}_t)$ | variational approximation, $\pi(\mathbf{z}_t) \approx p(\mathbf{z}_t \,|\, \mathbf{y}_{1:T})$ |
| $\boldsymbol{\lambda}_t$ | natural parameters of $\pi(\mathbf{z}_t)$ |
| $\boldsymbol{\mu}_t$ | mean parameters of $\pi(\mathbf{z}_t)$ |
| $\mathbf{m}_t$ / $\mathbf{P}_t$ | mean / covariance of $\pi(\mathbf{z}_t)$ |
| $\boldsymbol{\alpha}_t$ | local natural parameter update, $\boldsymbol{\alpha}_\phi(\mathbf{y}_t)$ |
| $\boldsymbol{\beta}_{t+1}$ | backward natural parameter update, $\boldsymbol{\beta}_\phi(\mathbf{y}_{t+1:T})$ |
| $\bar{\pi}(\mathbf{z}_t)$ | variational approximation with mean parameters $\bar{\boldsymbol{\mu}}_t = \mathbb{E}_{\pi_{t-1}}\left[\boldsymbol{\mu}_\theta(\mathbf{z}_{t-1})\right]$ |
| $\bar{\boldsymbol{\lambda}}_t$ | natural parameters of $\bar{\pi}(\mathbf{z}_t)$ |
| $\bar{\boldsymbol{\mu}}_t$ | mean parameters of $\bar{\pi}(\mathbf{z}_t)$ |
| $\bar{\mathbf{m}}_t$ / $\bar{\mathbf{P}}_t$ | mean / covariance of $\bar{\pi}(\mathbf{z}_t)$ |
| $\boldsymbol{\theta}$ | parameters of the dynamics and initial condition |
| $p_{\boldsymbol{\theta}}(\mathbf{z}_t \,|\, \mathbf{z}_{t-1})$ | prior over state transitions |
| $p_{\boldsymbol{\theta}}(\mathbf{z}_1)$ | prior over initial condition |
| $\boldsymbol{\psi}$ | parameters of the observation model/likelihood |
| $p_{\boldsymbol{\psi}}(\mathbf{y}_t \,|\, \mathbf{z}_t)$ | observation model/likelihood |

## B  Variational filtering

Principles of Bayesian inference make it straightforward to write down an algorithm recursively computing the filtering posterior, $p(\mathbf{z}_t \,|\, \mathbf{y}_{1:t})^2$. Given, $p(\mathbf{z}_{t-1} \,|\, \mathbf{y}_{1:t-1})$, updating our belief to $p(\mathbf{z}_t \,|\, \mathbf{y}_{1:t})$ after observing $\mathbf{y}_t$ can be broken down into two steps: first, we marginalize $p(\mathbf{z}_{t-1} \,|\, \mathbf{y}_{1:t-1})$ through the dynamics to obtain the predictive distribution,

$$\bar{p}(\mathbf{z}_t \,|\, \mathbf{y}_{1:t-1}) = \mathbb{E}_{p(\mathbf{z}_{t-1}|\mathbf{y}_{1:t-1})}\left[p_{\boldsymbol{\theta}}(\mathbf{z}_t \,|\, \mathbf{z}_{t-1})\right] \qquad \textbf{(predict step)} \tag{30}$$

Then, we update our belief by incorporating $\mathbf{y}_t$ through Bayes' rule,

$$p(\mathbf{z}_t \,|\, \mathbf{y}_{1:t}) \propto p_{\boldsymbol{\psi}}(\mathbf{y}_t \,|\, \mathbf{z}_t)\bar{p}(\mathbf{z}_t \,|\, \mathbf{y}_{1:t-1}) \qquad \textbf{(update step)} \tag{31}$$

With all of the filtered/predictive beliefs, the **smoothing step** is given by,

$$p(\mathbf{z}_t \,|\, \mathbf{y}_{1:T}) = \mathbb{E}_{p(\mathbf{z}_{t+1}|\mathbf{y}_{1:T})}\left[p(\mathbf{z}_t \,|\, \mathbf{z}_{t+1}, \mathbf{y}_{1:t})\right] = p(\mathbf{z}_t \,|\, \mathbf{y}_{1:t})\frac{\mathbb{E}_{p(\mathbf{z}_{t+1}|\mathbf{y}_{1:T})}\left[p_{\boldsymbol{\theta}}(\mathbf{z}_{t+1} \,|\, \mathbf{z}_t)\right]}{\mathbb{E}_{p(\mathbf{z}_t|\mathbf{y}_{1:t})}\left[p_{\boldsymbol{\theta}}(\mathbf{z}_{t+1} \,|\, \mathbf{z}_t)\right]} \tag{32}$$

However, these steps can usually not be evaluated in closed form when we depart from assumptions of Gaussianity and linearity. For nonlinear Gaussian dynamics the predict step can not be carried out exactly, and for nonlinear or non-Gaussian observations neither can the update step.

Alternatively, by considering variational analogues of the predict / update steps, we can develop a recursive and fully differentiable procedure for finding approximations $\pi(\mathbf{z}_t) \approx p(\mathbf{z}_t \,|\, \mathbf{y}_{1:t})$. In developing the variational analogues, it is assumed the approximations belong to an exponential family of distributions, (i.e. $\pi \in \mathcal{Q}$ where $\mathcal{Q}$ is an exponential family distribution) – not necessarily Gaussian.

### B.1  Variational predict step

Similar to developing a recursive algorithm as in the exact case, given $\pi(\mathbf{z}_{t-1}) \approx p(\mathbf{z}_{t-1} \,|\, \mathbf{y}_{1:t-1})$, we approximately marginalize $\pi(\mathbf{z}_{t-1})$ through the dynamics, by solving the following variational (forward KL / moment-matching) problem,

$$\bar{\pi}(\mathbf{z}_t) = \underset{\bar{\pi} \in \mathcal{Q}}{\operatorname{argmin}} \; \mathbb{D}_{\mathrm{KL}}\left(\mathbb{E}_{\pi(\mathbf{z}_{t-1})}\left[p_{\boldsymbol{\theta}}(\mathbf{z}_t \,|\, \mathbf{z}_{t-1})\right] \,\middle|\middle|\, \bar{\pi}(\mathbf{z}_t)\right) \tag{33}$$

So that if $p_{\boldsymbol{\theta}}(\mathbf{z}_t \,|\, \mathbf{z}_{t-1}) \in \mathcal{Q}$, the optimization problem is minimized when the mean parameters of $\bar{\pi}(\mathbf{z}_t)$, denoted $\bar{\boldsymbol{\mu}}_t$, are set to the expected mean parameter transformation under $\pi(\mathbf{z}_{t-1})$,

$$\bar{\boldsymbol{\mu}}_t = \mathbb{E}_{\pi(\mathbf{z}_{t-1})}\left[\boldsymbol{\mu}_\theta(\mathbf{z}_{t-1})\right] \qquad \textbf{(variational predict step)} \tag{34}$$

For a LGSSM with $p_{\boldsymbol{\theta}}(\mathbf{z}_t \,|\, \mathbf{z}_{t-1}) = \mathcal{N}(\mathbf{F}\mathbf{z}_{t-1}, \mathbf{Q})$, using the fact that,

$$\boldsymbol{\mu}_{\boldsymbol{\theta}}(\mathbf{z}_{t-1}) = \begin{bmatrix} \mathbf{F}\mathbf{z}_{t-1} \\ -\frac{1}{2}\left(\mathbf{F}\mathbf{z}_{t-1}\mathbf{z}_{t-1}^{\top}\mathbf{F}^{\top} + \mathbf{Q}_{\boldsymbol{\theta}}\right) \end{bmatrix} \tag{35}$$

means that if $\pi(\mathbf{z}_{t-1}) = \mathcal{N}(\mathbf{m}_{t-1}, \mathbf{P}_{t-1})$, setting the mean and variance of $\bar{\pi}(\mathbf{z}_t)$ to,

$$\bar{\mathbf{m}}_t = \mathbf{F}\mathbf{m}_{t-1} \tag{36}$$

$$\bar{\mathbf{P}}_t = \mathbf{F}\mathbf{P}_{t-1}\mathbf{F}^{\top} + \mathbf{Q}_{\boldsymbol{\theta}} \tag{37}$$

minimizes the forward KL objective, and reassuringly, recovers the familiar Kalman filter predict step.

## B.2 Variational update step

For the variational analogue of the filtering update step, we use $\bar{\pi}(\mathbf{z}_t)$ as a prior for the latest observation, $\mathbf{y}_t$, and solve the following variational (reverse KL) problem,

$$\pi(\mathbf{z}_t) = \underset{\pi \in \mathcal{Q}}{\arg\min} \; \mathbb{D}_{\mathrm{KL}}(\pi(\mathbf{z}_t) \| p_{\boldsymbol{\psi}}(\mathbf{y}_t \,|\, \mathbf{z}_t)\bar{\pi}(\mathbf{z}_t)) \tag{38}$$

If we denote the natural parameters of $\pi(\mathbf{z}_t)$ by $\boldsymbol{\lambda}_t$, then the optimal $\boldsymbol{\lambda}_t$ satisfy the implicit equation[33],

$$\boldsymbol{\lambda}_t = \nabla_{\boldsymbol{\mu}_t}\mathbb{E}_{\pi_t}\left[\log p_{\boldsymbol{\psi}}(\mathbf{y}_t \,|\, \mathbf{z}_t)\right] + \bar{\boldsymbol{\lambda}}_t \qquad \textbf{(variational update step)} \tag{39}$$

This usually requires an iterative optimization procedure, except when the likelihood is conjugate to $\pi(\mathbf{z}_t)$ in which case, the likelihood must take the following form with respect to $\mathbf{z}_t$,

$$p_{\boldsymbol{\psi}}(\mathbf{y}_t \,|\, \mathbf{z}_t) \propto \exp(\mathcal{T}(\mathbf{z}_t)^{\top}\tilde{\boldsymbol{\lambda}}_t) \tag{40}$$

so that, as expected, the natural parameters of the solution are given as Bayes' rule would suggest – by adding the data dependent update to the natural parameters of the prior so that,

$$\boldsymbol{\lambda}_t = \tilde{\boldsymbol{\lambda}}_t + \bar{\boldsymbol{\lambda}}_t \tag{41}$$

For a LGSSM with $p_{\boldsymbol{\psi}}(\mathbf{y}_t \,|\, \mathbf{z}_t) = \mathcal{N}(\mathbf{C}\mathbf{z}_t, \mathbf{R})$, this results in the following updates,

$$\mathbf{h}_t = \bar{\mathbf{h}}_t + \mathbf{C}^{\top}\mathbf{R}^{-1}\mathbf{y}_t \tag{42}$$

$$\mathbf{J}_t = \bar{\mathbf{J}}_t + \mathbf{C}^{\top}\mathbf{R}^{-1}\mathbf{C} \tag{43}$$

which reassuringly recover the information form of the Kalman filter update step[16].

## B.3 Variational smoothing step

Letting $\breve{q}(\mathbf{z}_t) \approx p(\mathbf{z}_t \,|\, \mathbf{y}_{1:t})$ and $q(\mathbf{z}_{t+1}) \approx p(\mathbf{z}_{t+1} \,|\, \mathbf{y}_{1:T})$, we can calculate the statistics of $q(\mathbf{z}_t)$ approximating the smoothed marginal posterior, by minimizing the following objective with respect to the marginal distribution, $q(\mathbf{z}_t)$,

$$\mathcal{L}_S(\breve{q}) = \mathbb{D}_{\mathrm{KL}}\big(\breve{q}(\mathbf{z}_t) \big\| \, \mathbb{E}_{\breve{q}(\mathbf{z}_{t+1})}\left[q(\mathbf{z}_t \,|\, \mathbf{z}_{t+1})\right]\big) \tag{44}$$

$$= \mathbb{D}_{\mathrm{KL}}(\breve{q}(\mathbf{z}_t) \| \, q(\mathbf{z}_t)) - \mathbb{E}_{\breve{q}(\mathbf{z}_t)}\left[\log \mathbb{E}_{\breve{q}(\mathbf{z}_{t+1})}\left[\frac{p_{\boldsymbol{\theta}}(\mathbf{z}_{t+1} \,|\, \mathbf{z}_t)}{\mathbb{E}_{q(\mathbf{z}_t)}\left[p_{\boldsymbol{\theta}}(\mathbf{z}_{t+1} \,|\, \mathbf{z}_t)\right]}\right]\right] \tag{45}$$

$$\approx \mathbb{D}_{\mathrm{KL}}(\breve{q}(\mathbf{z}_t) \| \, q(\mathbf{z}_t)) - \mathbb{E}_{\breve{q}(\mathbf{z}_t)}\left[\log \mathbb{E}_{\breve{q}(\mathbf{z}_{t+1})}\left[\frac{p_{\boldsymbol{\theta}}(\mathbf{z}_{t+1} \,|\, \mathbf{z}_t)}{\bar{q}(\mathbf{z}_{t+1})}\right]\right] := \widehat{\mathcal{L}}_S(\breve{q}) \tag{46}$$

taking natural gradients, we find that at a fixed point, the smoothed marginal posterior parameters satisfy the implicit relationship,

$$\boldsymbol{\lambda}_t = \breve{\boldsymbol{\lambda}}_t + \nabla_{\boldsymbol{\mu}_t}\mathbb{E}_{q(\mathbf{z}_t)}\left[A\left(\boldsymbol{\lambda}_{\boldsymbol{\theta}}(\mathbf{z}_t) + \boldsymbol{\lambda}_{t+1} - \bar{\boldsymbol{\lambda}}_{t+1}\right) - A\left(\boldsymbol{\lambda}_{\boldsymbol{\theta}}(\mathbf{z}_t)\right)\right] \tag{47}$$

## B.4 Efficiently sampling structured marginals

---

**Algorithm 2** Nonlinear variational filtering

---

**Input:** $\mathbf{k}_{1:T}$, $\mathbf{K}_{1:T}$
**for** $t = 1$ **to** $T$ **do**
$\quad \bar{\mathbf{m}}_t = S^{-1} \sum \mathbf{m}_{\boldsymbol{\theta}}(\mathbf{z}_{t-1}^s)$
$\quad \bar{\mathbf{M}}_t^c = S^{-1/2} \left[ \mathbf{m}_{\boldsymbol{\theta}}(\mathbf{z}_{t-1}^1) - \bar{\mathbf{m}}_t \cdots \mathbf{m}_{\boldsymbol{\theta}}(\mathbf{z}_{t-1}^S) - \bar{\mathbf{m}}_t \right]$
$\quad \bar{\boldsymbol{\Upsilon}}_t = \text{Cholesky}(\mathbf{I}_S + \bar{\mathbf{M}}_t^{c\top}\mathbf{Q}^{-1}\bar{\mathbf{M}}_t^c)^{-1}$
$\quad \bar{\mathbf{h}}_t = \bar{\mathbf{P}}_t^{-1}\bar{\mathbf{m}}_t \quad$ # using $[\mathbf{Q}, \bar{\mathbf{M}}_t^c, \bar{\boldsymbol{\Upsilon}}_t]$ and Eq. (50)
$\quad \boldsymbol{\Upsilon}_t = \text{Cholesky}(\mathbf{I}_r + \mathbf{K}_t^\top\bar{\mathbf{P}}_t\mathbf{K}_t)^{-1} \quad$ # using $[\mathbf{Q}, \bar{\mathbf{M}}_t^c]$ and Eq. (23)
$\quad \mathbf{h}_t = \bar{\mathbf{h}}_t + \mathbf{k}_t$
$\quad \mathbf{m}_t = \mathbf{P}_t\mathbf{h}_t \quad$ # using $[\mathbf{Q}, \bar{\mathbf{M}}_t^c, \mathbf{K}_t, \boldsymbol{\Upsilon}_t]$ and Eqs. (52) and (23)
$\quad \bar{\mathbf{w}}_t^s \sim \mathcal{N}(\mathbf{0}, \mathbf{I}_{L+S})$
$\quad \bar{\mathbf{z}}_t^s = \bar{\mathbf{P}}_t^{1/2}\bar{\mathbf{w}}_t^s \quad$ # using $[\mathbf{Q}, \bar{\mathbf{M}}_t^c]$ and Eq. (23)
$\quad \mathbf{w}_t^s \sim \mathcal{N}(\mathbf{0}, \mathbf{I}_r)$
$\quad \mathbf{z}_t^s = \mathbf{m}_t + \bar{\mathbf{z}}_t^s - \mathbf{K}_t\boldsymbol{\Upsilon}_t\boldsymbol{\Upsilon}_t^\top(\mathbf{K}_t^\top\bar{\mathbf{z}}_t^s + \mathbf{w}_t^s)$
**end for**
**Output:** $\mathbf{z}_{1:T}^{1:S}$, $\mathbf{m}_{1:T}$, $\bar{\mathbf{m}}_{1:T}$, $\boldsymbol{\Upsilon}_{1:T}$

---

While $\mathbf{P}_t$ has a structured representation, drawing samples from $\pi(\mathbf{z}_t)$ is not straightforward because we do not have a structured representation for a square-root of $\mathbf{P}_t$. However, using the factorization of $\bar{\mathbf{P}}_t$ in Eq. (23), it is possible to sample from $\mathcal{N}(\mathbf{0}, \bar{\mathbf{P}}_t)$ efficiently since $\bar{\mathbf{P}}_t^{1/2} = [\bar{\mathbf{M}}_t^c \ \mathbf{Q}^{1/2}]$. Now, combining the fact that the posterior marginal can be written as,

$$\pi(\mathbf{z}_t) = \mathcal{N}(\mathbf{m}_t, \mathbf{P}_t)$$
$$= \mathcal{N}(\mathbf{m}_t, (\bar{\mathbf{P}}_t^{-1} + \mathbf{K}_t\mathbf{K}_t^\top)^{-1}) \tag{48}$$

with the result from Cong et al. [34], stating that sampling $\mathbf{z}_t^s \sim \pi(\mathbf{z}_t)$ is equivalent to sampling $\mathbf{w}_t^s \sim \mathcal{N}(\mathbf{0}, \mathbf{I}_{L+S})$ and $\bar{\mathbf{z}}_t^s \sim \mathcal{N}(\mathbf{0}, \bar{\mathbf{P}}_t)$, and then setting

$$\mathbf{z}_t^s = \mathbf{m}_t + \bar{\mathbf{z}}_t^s - \mathbf{K}_t\boldsymbol{\Upsilon}_t\boldsymbol{\Upsilon}_t^\top(\mathbf{K}_t^\top\bar{\mathbf{z}}_t^s + \mathbf{w}_t^s), \tag{49}$$

makes it possible to efficiently draw samples from the posterior marginal.

## B.5 Efficient filtering

Evaluating $\bar{\mathbf{h}}_t = \bar{\mathbf{P}}_t^{-1}\bar{\mathbf{m}}_t$ and MVMs with $\bar{\mathbf{P}}_t^{-1}$, can be carried out in $\mathcal{O}(LS + S^2)$ time, after an initial cost of $\mathcal{O}(LS^2 + S^3)$ to factorize $\bar{\boldsymbol{\Upsilon}}_t\bar{\boldsymbol{\Upsilon}}_t^\top = (\mathbf{I}_S + \bar{\mathbf{M}}_t^{c\top}\mathbf{Q}_{\boldsymbol{\theta}}^{-1}\bar{\mathbf{M}}_t^c)^{-1}$, by applying the Woodbury identity to (23),

$$\bar{\mathbf{P}}_t^{-1} = \mathbf{Q}_{\boldsymbol{\theta}}^{-1} - \mathbf{Q}_{\boldsymbol{\theta}}^{-1}\bar{\mathbf{M}}_t^c\bar{\boldsymbol{\Upsilon}}_t\bar{\boldsymbol{\Upsilon}}_t^\top\bar{\mathbf{M}}_t^{c\top}\mathbf{Q}_{\boldsymbol{\theta}}^{-1} \tag{50}$$

Since this specifies all quantities that characterize $\bar{\pi}(\mathbf{z}_t)$, following (13), next is to update our belief by adding the information from the pseudo observation to them,

$$\mathbf{h}_t = \bar{\mathbf{h}}_t + \mathbf{k}_t \qquad\qquad \mathbf{P}_t^{-1} = \bar{\mathbf{P}}_t^{-1} + \mathbf{K}_t\mathbf{K}_t^\top \tag{51}$$

As a result, the multiplication in $\mathbf{m}_t = \mathbf{P}_t\mathbf{h}_t$ requires $\mathcal{O}(LS + Lr)$ time, and, analogous to (50), the square root factor $\boldsymbol{\Upsilon}_t$ in

$$\mathbf{P}_t = \bar{\mathbf{P}}_t - \bar{\mathbf{P}}_t\mathbf{K}_t\boldsymbol{\Upsilon}_t\boldsymbol{\Upsilon}_t^\top\mathbf{K}_t^\top\bar{\mathbf{P}}_t \tag{52}$$

requires $\mathcal{O}(r^3 + LSr + S^2r)$ time where $\boldsymbol{\Upsilon}_t\boldsymbol{\Upsilon}_t^\top = (\mathbf{I}_r + \mathbf{K}_t^\top\bar{\mathbf{P}}_t\mathbf{K}_t)^{-1}$.

# C Evaluating the KL using low-rank structure

## C.1 Smoothing inference network

Efficient training of the generative model and inference networks require efficient numerical evaluation of the ELBO. We take advantage of the structured precision matrices arising from low-rank updates

and sample approximations. The expected log likelihood can be evaluated using a Monte Carlo approximation from samples during the filtering pass. The KL term,

$$\mathbb{D}_{\text{KL}}(\pi(\mathbf{z}_t) \| \bar{\pi}(\mathbf{z}_t)) = \tfrac{1}{2} \left[ (\bar{\mathbf{m}}_t - \mathbf{m}_t)^\top \bar{\mathbf{P}}_t^{-1} (\bar{\mathbf{m}}_t - \mathbf{m}_t) + \text{tr}(\bar{\mathbf{P}}_t^{-1} \mathbf{P}_t) + \log \frac{|\bar{\mathbf{P}}_t|}{|\mathbf{P}_t|} - L \right] \quad (53)$$

can be evaluated in closed form and we can expand each term as,

**Log-determinant.** Writing

$$\log |\mathbf{P}_t| = - \log |\bar{\mathbf{P}}_t^{-1} + \mathbf{K}_t \mathbf{K}_t^\top| \quad (54)$$

$$= \log |\bar{\mathbf{P}}_t| - \log |\mathbf{I} + \mathbf{K}_t^\top \bar{\mathbf{P}}_t \mathbf{K}_t| \quad (55)$$

gives

$$\log \frac{|\bar{\mathbf{P}}_t|}{|\mathbf{P}_t|} = \log |\mathbf{I} + \mathbf{K}_t^\top \bar{\mathbf{P}}_t \mathbf{K}_t| \quad (56)$$

$$= -2 \sum_{i=1}^{r} [\mathbf{\Upsilon}_t]_{i,i} \quad (57)$$

**Trace.** Writing,

$$\text{tr}(\bar{\mathbf{P}}_t^{-1} \mathbf{P}_t) = \text{tr}(\mathbf{I}_L - \mathbf{K}_t (\mathbf{I} + \mathbf{K}_t^\top \bar{\mathbf{P}}_t \mathbf{K}_t)^{-1} \mathbf{K}_t^\top \bar{\mathbf{P}}_t) \quad (58)$$

$$= L - \text{tr}(\bar{\mathbf{P}}_t^{\top/2} \mathbf{K}_t (\mathbf{I} + \mathbf{K}_t^\top \bar{\mathbf{P}}_t \mathbf{K}_t)^{-1} \mathbf{K}_t^\top \bar{\mathbf{P}}_t^{1/2}) \quad (59)$$

$$= L - \text{tr}(\mathbf{\Upsilon}_t^\top \mathbf{K}_t^\top \bar{\mathbf{P}}_t \mathbf{K}_t \mathbf{\Upsilon}_t) \quad (60)$$

which by taking $\mathbf{\Upsilon}_t$ to be an $r \times r$ square root such that,

$$\mathbf{\Upsilon}_t \mathbf{\Upsilon}_t^\top = (\mathbf{I} + \mathbf{K}_t^\top \bar{\mathbf{P}}_t \mathbf{K}_t)^{-1} \quad (61)$$

further simplifies to

$$\text{tr}(\bar{\mathbf{P}}_t^{-1} \mathbf{P}_t) = L - \text{tr}(\bar{\mathbf{M}}_t^{c\top} \mathbf{K}_t \mathbf{\Upsilon}_t \mathbf{\Upsilon}_t^\top \mathbf{K}_t^\top \bar{\mathbf{M}}_t^c) - \text{tr}(\mathbf{Q}^{\top/2} \mathbf{K}_t \mathbf{\Upsilon}_t \mathbf{\Upsilon}_t^\top \mathbf{K}_t^\top \mathbf{Q}^{1/2}) \quad (62)$$

$$= L - \text{tr}(\bar{\mathbf{M}}_t^{c\top} \mathbf{K}_t \mathbf{\Upsilon}_t \mathbf{\Upsilon}_t^\top \mathbf{K}_t^\top \bar{\mathbf{M}}_t^c) - \text{tr}(\mathbf{\Upsilon}_t^\top \mathbf{K}_t^\top \mathbf{Q}^{1/2} \mathbf{Q}^{\top/2} \mathbf{K}_t \mathbf{\Upsilon}_t) \quad (63)$$

note that the size of the triple product, $\bar{\mathbf{M}}_t^{c\top} \mathbf{K}_t \mathbf{\Upsilon}_t$, is $S \times r$.

## C.2 Causal/streaming inference network

When the real-time parameterization of the inference network is used the expressions become slightly more complicated due to the more intricate relationship between the posterior at time $t$ and the posterior predictive at time $t$.

**Log-determinant.** We need to first find $\log |\bar{\mathbf{P}}_{t|T}| - \log |\mathbf{P}_t|$ so begin with using the matrix-determinant lemma to write,

$$|\bar{\mathbf{P}}_{t|T}| = |\mathbf{M}_{t|T}^c \mathbf{M}_{t|T}^{c\top} + \mathbf{Q}| \quad (64)$$

$$= |\mathbf{I}_S + \mathbf{M}_{t|T}^{c\top} \mathbf{Q}^{-1} \mathbf{M}_{t|T}^c| \times |\mathbf{Q}| \quad (65)$$

then expand the smoothed covariance to write,

$$|\mathbf{P}_t| = |(\breve{\mathbf{P}}_t^{-1} + \mathbf{B}_t \mathbf{B}_t^\top)^{-1}| \quad (66)$$

$$= |\breve{\mathbf{P}}_t^{-1} + \mathbf{B}_t \mathbf{B}_t^\top|^{-1} \quad (67)$$

$$= (|\mathbf{I}_{r_\beta} + \mathbf{B}_t^\top \breve{\mathbf{P}}_t \mathbf{B}_t| \times |\breve{\mathbf{P}}_t^{-1}|)^{-1} \quad (68)$$

$$= |\mathbf{I}_{r_\beta} + \mathbf{B}_t^\top \breve{\mathbf{P}}_t \mathbf{B}_t|^{-1} |\breve{\mathbf{P}}_t| \quad (69)$$

and another time to write,

$$|\breve{\mathbf{P}}_t^{-1}| = |\bar{\mathbf{P}}_t^{-1} + \mathbf{A}_t \mathbf{A}_t^\top| \quad (70)$$

$$= |\mathbf{I}_{r_\alpha} + \mathbf{A}_t^\top \bar{\mathbf{P}}_t \mathbf{A}_t| \times |\bar{\mathbf{P}}_t^{-1}| \quad (71)$$

and another time,

$$|\bar{\mathbf{P}}_t| = |\mathbf{M}_t^c \mathbf{M}_t^{c\top} + \mathbf{Q}| \tag{72}$$

$$= |\mathbf{I}_S + \mathbf{M}_t^{c\top} \mathbf{Q}^{-1} \mathbf{M}_t^c| \times |\mathbf{Q}| \tag{73}$$

When combined we are finally able to write

$$\log|\bar{\mathbf{P}}_{t|T}| - \log|\mathbf{P}_t| = \log|\mathbf{I}_S + \mathbf{M}_{t|T}^{c\top} \mathbf{Q}^{-1} \mathbf{M}_{t|T}^c| + \log|\mathbf{I}_{r_\beta} + \mathbf{B}_t^\top \breve{\mathbf{P}}_t \mathbf{B}_t| \tag{74}$$

$$+ \log|\mathbf{I}_{r_\alpha} + \mathbf{A}_t^\top \bar{\mathbf{P}}_t \mathbf{A}_t| - \log|\mathbf{I}_S + \mathbf{M}_t^{c\top} \mathbf{Q}^{-1} \mathbf{M}_t^c| \tag{75}$$

For the initial condition we have,

$$\log|\bar{\mathbf{P}}_1| - \log|\mathbf{P}_1| = \log|\mathbf{I}_{r_\beta} + \mathbf{B}_1^\top \breve{\mathbf{P}}_1 \mathbf{B}_1| + \log|\mathbf{I}_{r_\alpha} + \mathbf{A}_1^\top \bar{\mathbf{P}}_1 \mathbf{A}_1| \tag{76}$$

**Trace.** For the trace,

$$\mathrm{tr}(\bar{\mathbf{P}}_{t|T}^{-1} \mathbf{P}_t) = \mathrm{tr}(\mathbf{Q}^{-1} \mathbf{P}_t) - \mathrm{tr}(\mathbf{Q}^{-1} \mathbf{M}_{t|T}^c (\mathbf{I}_S + \mathbf{M}_{t|T}^{c\top} \mathbf{Q}^{-1} \mathbf{M}_{t|T}^c)^{-1} \mathbf{M}_{t|T}^{c\top} \mathbf{Q}^{-1} \mathbf{P}_t) \tag{77}$$

$$= \mathrm{tr}(\mathbf{Q}^{-1} \mathbf{P}_t) - \mathrm{tr}(\mathbf{Q}^{-1} \mathbf{M}_{t|T}^c \bar{\boldsymbol{\Upsilon}}_{t|T} \bar{\boldsymbol{\Upsilon}}_{t|T}^\top \mathbf{M}_{t|T}^{c\top} \mathbf{Q}^{-1} \mathbf{P}_t) \tag{78}$$

$$= \mathrm{tr}(\mathbf{P}_t \mathbf{Q}^{-1}) - \mathrm{tr}([\mathbf{P}_t \mathbf{Q}^{-1} \mathbf{M}_{t|T}^c \bar{\boldsymbol{\Upsilon}}_{t|T}] \bar{\boldsymbol{\Upsilon}}_{t|T}^\top \mathbf{M}_{t|T}^{c\top} \mathbf{Q}^{-1}) \tag{79}$$

where we expand the first rhs term for a numerically efficient implementation by writing

$$\mathrm{tr}(\mathbf{P}_t \mathbf{Q}^{-1}) = \mathrm{tr}(\bar{\mathbf{P}}_t \mathbf{Q}^{-1}) - \mathrm{tr}(\bar{\mathbf{P}}_t \mathbf{A}_t (\mathbf{I}_{r_\alpha} + \mathbf{A}_t^\top \bar{\mathbf{P}}_t \mathbf{A}_t)^{-1} \mathbf{A}_t^\top \bar{\mathbf{P}}_t \mathbf{Q}^{-1}) \tag{80}$$

$$- \mathrm{tr}(\breve{\mathbf{P}}_t \mathbf{B}_t (\mathbf{I}_{r_\beta} + \mathbf{B}_t^\top \breve{\mathbf{P}}_t \mathbf{B}_t)^{-1} \mathbf{B}_t^\top \breve{\mathbf{P}}_t \mathbf{Q}^{-1}) \tag{81}$$

To reduce notational clutter we define,

$$\bar{\boldsymbol{\Upsilon}}_t \bar{\boldsymbol{\Upsilon}}_t^\top = (\mathbf{I}_S + \mathbf{M}_t^{c\top} \mathbf{Q}^{-1} \mathbf{M}_t^c)^{-1} \tag{82}$$

$$\bar{\boldsymbol{\Upsilon}}_{t|T} \bar{\boldsymbol{\Upsilon}}_{t|T}^\top = (\mathbf{I}_S + \mathbf{M}_{t|T}^{c\top} \mathbf{Q}^{-1} \mathbf{M}_{t|T}^c)^{-1} \tag{83}$$

$$\breve{\boldsymbol{\Upsilon}}_t \breve{\boldsymbol{\Upsilon}}_t^\top = (\mathbf{I}_{r_\alpha} + \mathbf{A}_t^\top \breve{\mathbf{P}}_t \mathbf{A}_t)^{-1} \tag{84}$$

$$\boldsymbol{\Upsilon}_t \boldsymbol{\Upsilon}_t^\top = (\mathbf{I}_{r_\beta} + \mathbf{B}_t^\top \bar{\mathbf{P}}_t \mathbf{B}_t)^{-1} \tag{85}$$

and so the trace is

$$\mathrm{tr}(\bar{\mathbf{P}}_{t|T}^{-1} \mathbf{P}_t) = L + \mathrm{tr}(\mathbf{M}_t^{c\top} \mathbf{Q}^{-1} \mathbf{M}_t^c) - \mathrm{tr}(\mathbf{Q}^{-1/2} \bar{\mathbf{P}}_t \mathbf{A}_t \breve{\boldsymbol{\Upsilon}}_t \breve{\boldsymbol{\Upsilon}}_t^\top \mathbf{A}_t^\top \bar{\mathbf{P}}_t \mathbf{Q}^{-1/2}) \tag{86}$$

$$- \mathrm{tr}(\mathbf{Q}^{-1/2} \breve{\mathbf{P}}_t \mathbf{B}_t \boldsymbol{\Upsilon}_t \boldsymbol{\Upsilon}_t^\top \mathbf{B}_t^\top \breve{\mathbf{P}}_t \mathbf{Q}^{-1/2}) \tag{87}$$

$$- \mathrm{tr}([\mathbf{P}_t \mathbf{Q}^{-1} \mathbf{M}_{t|T}^c \bar{\boldsymbol{\Upsilon}}_{t|T}] \bar{\boldsymbol{\Upsilon}}_{t|T}^\top \mathbf{M}_{t|T}^{c\top} \mathbf{Q}^{-1}) \tag{88}$$

which is now in a form that is easy to handle using fast MVMs with $\bar{\mathbf{P}}_t, \breve{\mathbf{P}}_t, \mathbf{P}_t$.

For the initial condition term we use the fact that $\bar{\mathbf{P}}_1$ is diagonal,

$$\mathrm{tr}(\mathbf{P}_1 \bar{\mathbf{P}}_1^{-1}) = L - \mathrm{tr}(\bar{\mathbf{P}}_1^{1/2} \mathbf{A}_1 \breve{\boldsymbol{\Upsilon}}_1 \breve{\boldsymbol{\Upsilon}}_1^\top \mathbf{A}_1^\top \bar{\mathbf{P}}_1^{1/2}) - \mathrm{tr}(\bar{\mathbf{P}}_1^{-1/2} \breve{\mathbf{P}}_1 \mathbf{B}_1 \boldsymbol{\Upsilon}_1 \boldsymbol{\Upsilon}_1^\top \mathbf{B}_1^\top \breve{\mathbf{P}}_1 \bar{\mathbf{P}}_1^{-1/2}) \tag{89}$$

## D  Comparison method details

### D.1  SVAE

For the SVAE[20], the latent state prior is a linear dynamical system parameterized as,

$$p_{\boldsymbol{\theta}}(\mathbf{z}_t | \mathbf{z}_{t-1}) = \mathcal{N}(\mathbf{z}_t | \mathbf{F}\mathbf{z}_{t-1}, \mathbf{Q}) \tag{90}$$

Using conjugate potentials, with likelihood $p(\tilde{\mathbf{y}}_t | \mathbf{z}_t) = \exp(t(\mathbf{z}_t)^\top \boldsymbol{\alpha}(\mathbf{y}_t))$, the approximate posterior is given by $q(\mathbf{z}_{1:T}) = \prod p(\tilde{\mathbf{y}}_t | \mathbf{z}_t) p_{\boldsymbol{\theta}}(\mathbf{z}_{1:T})$ so that its statistics can be found by applying Kalman filtering/smoothing to the pseudo-observations. In this case, the ELBO can be evaluated as

$$\mathcal{L}(q) = \sum \mathbb{E}_{q_t} \left[\log p(\mathbf{y}_t | \mathbf{z}_t)\right] - \mathbb{E}_{q_t} \left[\log p(\tilde{\mathbf{y}}_t | \mathbf{z}_t)\right] + \log \mathbb{E}_{\bar{q}_t} \left[p(\tilde{\mathbf{y}}_t | \mathbf{z}_t)\right] \tag{91}$$

where $\bar{q}_t := \bar{q}(\mathbf{z}_t) = p(\mathbf{z}_t | \tilde{\mathbf{y}}_{1:t-1})$ is the filtering predictive distribution. These expressions can be evaluated in closed form and written concisely in terms of natural/mean parameters and log-partition functions as

$$\mathcal{L}(q) = \sum \mathbb{E}_{q_t} \left[\log p(\mathbf{y}_t | \mathbf{z}_t)\right] - \boldsymbol{\mu}_t^\top \boldsymbol{\alpha}_t + A(\bar{\boldsymbol{\lambda}}_t + \boldsymbol{\alpha}_t) - A(\bar{\boldsymbol{\lambda}}_t) \tag{92}$$

where $\boldsymbol{\mu}_t = \nabla A(\bar{\boldsymbol{\lambda}}_t + \boldsymbol{\alpha}_t + \boldsymbol{\beta}_{t+1})$. Using the identities given in App. F, these expressions can be written in more familiar mean/covariance parameters.

## D.2 DVBF

For the DVBF[5], we parameterize the latent state prior using a nonlinear dynamical system of the same form as Eq. (1). Then, using an inference network that encodes data in reverse-time to produce the parameters of a diagonal Gaussian distribution, $\mathbf{w}_t \sim q(\mathbf{w}_t) = \mathcal{N}(\mathbf{m}_t, \mathrm{diag}(\mathbf{s}_t))$, we sample the latent trajectory forward using the recursion, $\mathbf{z}_t = \mathbf{m}_{\boldsymbol{\theta}}(\mathbf{z}_{t-1}) + \mathbf{Q}^{1/2}\mathbf{w}_t$. Parameters of the generative model/inference network are learned jointly by minimizing the ELBO,

$$\mathcal{L}(q) = \sum \mathbb{E}_{q(\mathbf{z}_t)}\left[\log p_{\boldsymbol{\psi}}(\mathbf{y}_t \mid \mathbf{z}_t)\right] - \mathbb{D}_{\mathrm{KL}}(q(\mathbf{w}_t)\|\,p(\mathbf{w}_t)) \tag{93}$$

where, $p(\mathbf{w}_t) = \mathcal{N}(\mathbf{0}, \mathbf{I})$.

## D.3 DKF

For the DKF[6], the latent state is also parameterized using a nonlinear dynamical system of the same form as Eq. (1). We follow the parameterization outlined in the text with S2S$(\cdot)$ implemented using a recurrent neural network. We sample trajectories using the inference network and jointly train all parameters on the ELBO,

$$\mathcal{L}(q) = \sum \mathbb{E}_{q_t}\left[\log p(\mathbf{y}_t \mid \mathbf{z}_t)\right] - \mathbb{E}_{q_{t-1}}\left[\mathbb{D}_{\mathrm{KL}}(q(\mathbf{z}_t \mid \mathbf{z}_{t-1})\|\,p_{\boldsymbol{\theta}}(\mathbf{z}_t \mid \mathbf{z}_{t-1}))\right] \le \log p(\mathbf{y}_{1:T}) \tag{94}$$

# E Experimental details

In describing the neural network architectures used to parameterize the inference model, it will be useful to define the following multilayer perceptron (MLP), with SiLU nonlinearity[35], that gets used repeatedly:

– $\mathrm{MLP}(n_{\mathrm{in}}, n_{\mathrm{hidden}}, n_{\mathrm{out}}) : [\mathrm{Linear}(n_{\mathrm{in}}, n_{\mathrm{hidden}}), \mathrm{SiLU}(), \mathrm{Linear}(n_{\mathrm{hidden}}, n_{\mathrm{out}})]$

$$\mathcal{L}(q) = \sum \mathbb{E}_{q_t}\left[\log p(\mathbf{y}_t \mid \mathbf{z}_t)\right] - \mathbb{E}_{q_{t-1}}\left[\mathbb{D}_{\mathrm{KL}}(q(\mathbf{z}_t \mid \mathbf{z}_{t-1})\|\,p_{\boldsymbol{\theta}}(\mathbf{z}_t \mid \mathbf{z}_{t-1}))\right] \le \log p(\mathbf{y}_{1:T}) \tag{95}$$

## E.1 High-dimensional linear dynamical system

We simulated data from an LDS generative model (with dynamics restricted to the set of matrices with singular values less than 1) for latent dimensions $L \in [20, 50, 100]$ over 3 random seeds for two scenarios **i)** $N = L$ and **ii)** $N = L/5$. For each scenario, we also vary the rank of the local/backward encoder precision updates.

## E.2 Time complexity

We generate random LDS systems of appropriate dimension and measure the average time to complete one forward pass and take a gradient step. The system used for benchmarking wall-clock time was an RTX 4090 with 128GB of RAM with an AMD 5975WX processor.

## E.3 Pendulum

We consider the pendulum system from[26]. We generate 500/150/150 trials of length 100 for training/validation/testing. All methods are trained for 5000 epochs for 3 different random seeds. We consider a context window of 50 images and a forecast window of 50 images. A decoder was fit from the latent state on the training set during the context window; then, for held out data, we examine performance of the decoder during the context and forecast windows.

The generative model is parameterized as:

- $L = 4$
- $N = 256$
- $T = 50$
- likelihood

- $p_\psi(\mathbf{y}_t | \mathbf{z}_t) = \mathcal{N}(\mathbf{z}_t | \mathbf{C}_\psi(\mathbf{z}_t) + \mathbf{b}, \mathbf{R})$
  - $\mathbf{C}_\psi : \mathrm{MLP}(4, 128, 256)$

For each method, inference is amortized using the following neural network architectures:

- our inference network
  - $r_\alpha = 4$
  - $r_\beta = 4$
  - $\boldsymbol{\alpha}_\phi$: $\mathrm{MLP}(256, 128, 20)$
  - $\boldsymbol{\beta}_\phi$: $[\mathrm{GRU}(128), \mathrm{Linear}(128, 20)]$
- DKF inference network
  - $\mathbf{u}_\phi$: $\mathrm{GRU}(128)$
  - $(\mathbf{m}_\phi, \log \mathbf{P}_\phi)$: $\mathrm{MLP}(132, 128, 8)$
- DVBF inference network
  - $\mathbf{u}_\phi$: $[\mathrm{GRU}(128)]$
  - $(\boldsymbol{\mu}_\phi, \log \boldsymbol{\sigma}_\phi)$: $\mathrm{MLP}(132, 128, 8)$
- SVAE inference network
  - $\boldsymbol{\alpha}_\phi$: $\mathrm{MLP}(256, 128, 20)$

Optimization and training details:

- optimizer: $\mathrm{Adam}(\mathrm{lr} = 0.001)$
- batch size: 128

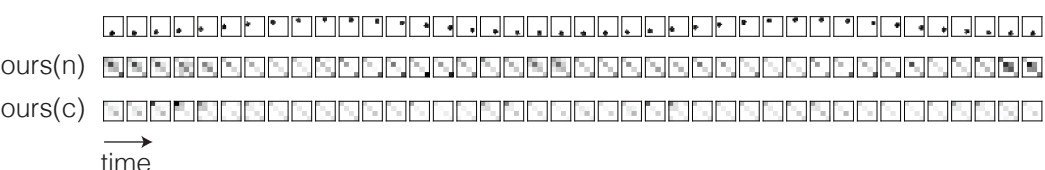

Figure 4: **Learned covariance of nonlinear SSMs**. (top) single trial of a pendulum. Below are the posterior covariances output by the causal and non-causal variants of XFADS.

### E.4 Bouncing ball

We consider a bouncing ball dataset commonly used as a baseline to benchmark the performance of inference and learning in deep state-space models [26,36,36]. For this dataset we take $500/150/150$ trials of length 75 for training/validation/testing. All methods are trained for 5000 epochs for 3 different random seeds. The generative model is parameterized as:

- $L = 8$
- $N = 256$
- $T = 50$
- likelihood
  - $p_\psi(\mathbf{y}_t | \mathbf{z}_t) = \mathcal{N}(\mathbf{z}_t | \mathbf{C}_\psi(\mathbf{z}_t) + \mathbf{b}, \mathbf{R})$
  - $\mathbf{C}_\psi : \mathrm{MLP}(8, 128, 256)$

For each method, inference is amortized using the following neural network architectures:

- our inference network
  - $r_\alpha = 8$
  - $r_\beta = 4$
  - $\boldsymbol{\alpha}_\phi$: $\mathrm{MLP}(256, 128, 72)$

- $\boldsymbol{\beta}_\phi$: $[\mathrm{GRU}(128), \mathrm{Linear}(128, 40)]$
- DKF inference network
  - $\mathbf{u}_\phi$: $\mathrm{GRU}(128)$
  - $(\mathbf{m}_\phi, \log\mathbf{P}_\phi)$: $\mathrm{MLP}(132, 128, 16)$
- DVBF inference network
  - $\mathbf{u}_\phi$: $[\mathrm{GRU}(128)]$
  - $(\boldsymbol{\mu}_\phi, \log\boldsymbol{\sigma}_\phi)$: $\mathrm{MLP}(132, 128, 16)$
- SVAE inference network
  - $\boldsymbol{\alpha}_\phi$: $\mathrm{MLP}(256, 128, 72)$

Optimization and training details:

- optimizer: $\mathrm{Adam}(\mathrm{lr} = 0.001)$
- batch size: 128

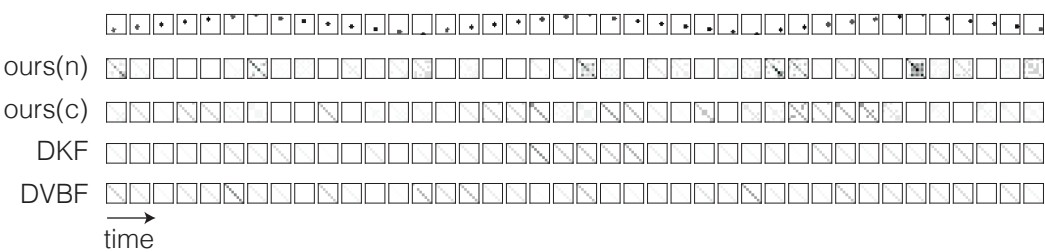

Figure 5: **Learned covariance of nonlinear SSMs**. (top) single trial of a bouncing ball. Below are the posterior covariances output by the nonlinear SSMs considered – qualitatively, we observe more complex covariance structures arise when the ball hits the wall that diagonal approximations cannot capture.

### E.5 MC_Maze

The monkey reaching dataset of[30] was the first real dataset examined in the main text. For this dataset, we partitioned 1800/200/200 training/validation/testing trials sampled at 20ms per bin. All methods are trained for 1000 epochs for 3 different random seeds. The generative model is parameterized as:

- $L = 40$
- $N = 182$
- $T = 35$
- likelihood
  - $p_\psi(\mathbf{y}_t | \mathbf{z}_t) = \mathrm{Poisson}(\mathbf{z}_t | \mathbf{C}_\psi(\mathbf{z}_t) + \mathbf{b})$
  - $\mathbf{C}_\psi$ : $\mathrm{Linear}(40, 182)$

For each method, inference is amortized using the following neural network architectures:

- our inference network
  - $r_\alpha = 15$
  - $r_\beta = 5$
  - $\boldsymbol{\alpha}_\phi$: $\mathrm{MLP}(182, 128, 640)$
  - $\boldsymbol{\beta}_\phi$: $[\mathrm{GRU}(128), \mathrm{Linear}(128, 240)]$
- DKF inference network
  - $\mathbf{u}_\phi$: $\mathrm{GRU}(128)$
  - $(\mathbf{m}_\phi, \log\mathbf{P}_\phi)$: $\mathrm{MLP}(128, 128, 80)$

- DVBF inference network
    - $\mathbf{u}_\phi$: $[\text{GRU}(128)]$
    - $(\mathbf{m}_\phi, \log \mathbf{P}_\phi)$: $\text{MLP}(128, 128, 80)$
- SVAE inference network
    - $\boldsymbol{\alpha}_\phi$: $\text{MLP}(182, 256, 1640)$

Optimization and training details:

- optimizer: $\text{Adam}(\text{lr} = 0.001)$
- batch size: $128$

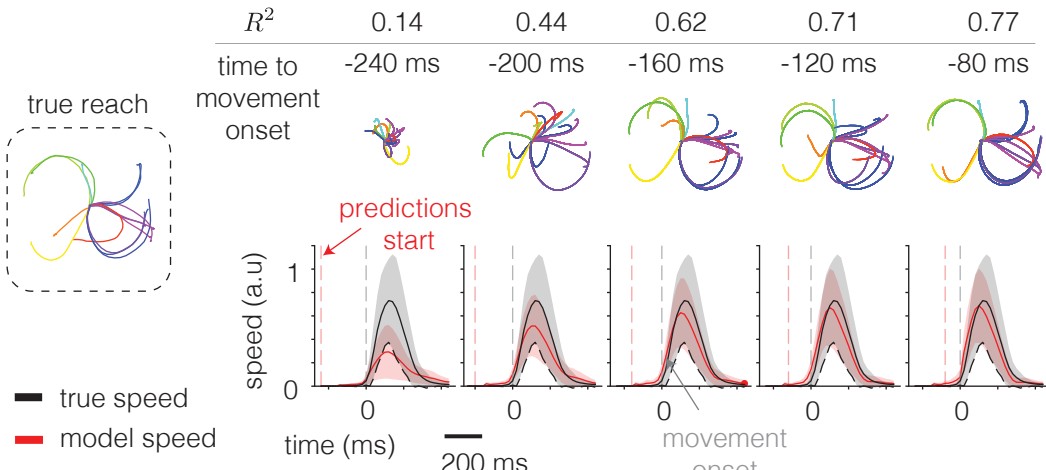

Figure 6: **XFADS predictive capabilities on real data.** On the left are example reaches the monkey made during several trials. Using the learned model from the monkey reaching experiment, we filtered neural activity starting from $-240$ms up until $-80$ms before movement onset (each vertical panel represents the result from filtering more and more data) and unroll the final filtered states through the dynamics (starting from the dashed red line) to make predictions.

## E.6  DMFC_RSG

The second real dataset examined was the timing interval reproduction task of[32] samples at 10ms bins. For this dataset, we partitioned 700/150/150 training/validation/testing trials. All methods are trained for 1000 epochs for 3 different random seeds. The generative model is parameterized as:

- $L = 40$
- $N = 54$
- $T = 130$
- likelihood
    - $p_\psi(\mathbf{y}_t | \mathbf{z}_t) = \text{Poisson}(\mathbf{z}_t | \mathbf{C}_\psi(\mathbf{z}_t) + \mathbf{b})$
    - $\mathbf{C}_\psi : \text{Linear}(40, 54)$

For each method, inference is amortized using the following neural network architectures:

- our inference network
    - $r_\alpha = 15$
    - $r_\beta = 5$
    - $\boldsymbol{\alpha}_\phi$: $\text{MLP}(54, 128, 640)$
    - $\boldsymbol{\beta}_\phi$: $[\text{GRU}(128), \text{Linear}(128, 240)]$

- DKF inference network
    - $\mathbf{u}_\phi$: GRU(128)
    - $(\mathbf{m}_\phi, \log \mathbf{P}_\phi)$: MLP(128, 128, 80)
- DVBF inference network
    - $\mathbf{u}_\phi$: [GRU(128)]
    - $(\boldsymbol{\mu}_\phi, \log \boldsymbol{\sigma}_\phi)$: MLP(128, 128, 80)
- SVAE inference network
    - $\boldsymbol{\alpha}_\phi$: MLP(256, 128, 1640)

Optimization and training details:

- optimizer: $\mathrm{Adam}(\mathrm{lr} = 0.001)$
- batch size: 128

### E.7 Collated results

| | pendulum | | bouncing ball | | monkey reaching | |
|---|---|---|---|---|---|---|
| | smoothing | prediction | smoothing | prediction | smoothing | prediction |
| ours (n) | 99.7 (.01) | 72.7 (3.4) | 80.7 (.70) | 34.6 (1.5) | 88.5 (.1) | 52.7 (1.7) |
| ours (c) | 99.6 (.01) | 69.7 (3.0) | 80.4 (.60) | 29.1 (4.7) | 87.8 (.1) | 53.3 (1.1) |
| DKF | 99.2 (.70) | 19.1 (9.5) | 65.6 (4.9) | 0.32 (10.) | 81.2 (.3) | 1.7 (9.1) |
| DVBF | 90.7 (2.8) | -19.6 (9.8) | 75.3 (.70) | 1.3 (.20) | 86.3 (.1) | 33.4 (.5) |
| SVAE | 98.4 (.40) | -39.7 (10.) | 76.5 (2.1) | -23.3 (.70) | 87.5 (.1) | -2.4 (13.) |
| L-SDE | 92.1 (2.3) | 13.8 (12.) | 81.3 (.70) | 23.1 (1.3) | 76.6 (.4) | 18.7 (1.0) |
| LFADS | | | | | 91.1 (.1) | 23.2 (1.1) |

Figure 7: **Collated results including the latent SDE (L-SDE)**[37]. We ran an additional baseline on pendulum/bouncing ball/monkey reaching datasets. L-SDE achieves results with a similar level of performance as the other models. Listed are the $R^2$ values for decoding angular velocity/x-y position/reach velocity from the latent representations learned for each dataset in both smoothing and prediction regimes.

## F  Useful expressions

**mean/natural parameter inner product**   One common expression that frequently arises is the inner product between a mean and natural parameter, i.e.

$$\boldsymbol{\mu}_t^\top \boldsymbol{\alpha}_t \tag{96}$$

where in the Gaussian case if the mean/natural parameter coordinates are,

$$\boldsymbol{\mu}_t = \begin{pmatrix} \mathbf{m}_t \\ -\frac{1}{2}(\mathbf{P}_t + \mathbf{m}_t \mathbf{m}_t^\top) \end{pmatrix} \qquad\qquad \boldsymbol{\alpha}_t = \begin{pmatrix} \mathbf{a}_t \\ \mathbf{A}_t \mathbf{A}_t^\top \end{pmatrix} \tag{97}$$

then,

$$\boldsymbol{\mu}_t^\top \boldsymbol{\alpha}_t = \mathbf{m}_t^\top \mathbf{a}_t - \frac{1}{2}||\mathbf{A}_t^\top \mathbf{m}_t||^2 - \frac{1}{2}\operatorname{tr}(\mathbf{A}_t^\top \mathbf{P}_t \mathbf{A}_t) \tag{98}$$

**difference of log partition functions**  Another common expression that frequently arises is given by,

$$A(\bar{\boldsymbol{\lambda}}_t + \boldsymbol{\alpha}_t) - A(\bar{\boldsymbol{\lambda}}_t) \tag{99}$$

so that if,

$$\bar{\boldsymbol{\lambda}}_t = \begin{pmatrix} \bar{\mathbf{P}}_t^{-1}\bar{\mathbf{m}}_t \\ \bar{\mathbf{P}}_t^{-1} \end{pmatrix} \tag{100}$$

then,

$$A(\bar{\boldsymbol{\lambda}}_t + \boldsymbol{\alpha}_t) - A(\bar{\boldsymbol{\lambda}}_t) \tag{101}$$

$$= \tfrac{1}{2}\breve{\mathbf{m}}_t^\top(\bar{\mathbf{P}}_t^{-1} + \mathbf{A}_t\mathbf{A}_t^\top)\breve{\mathbf{m}}_t - \tfrac{1}{2}\log|\bar{\mathbf{P}}_t^{-1} + \mathbf{A}_t\mathbf{A}_t^\top| \tag{102}$$

$$\quad - \tfrac{1}{2}\bar{\mathbf{m}}_t\bar{\mathbf{P}}_t^{-1}\bar{\mathbf{m}}_t + \tfrac{1}{2}\log|\bar{\mathbf{P}}_t^{-1}|$$

$$= \tfrac{1}{2}\left( ||\breve{\mathbf{m}}_t||_{\bar{\mathbf{P}}_t^{-1}}^2 - ||\bar{\mathbf{m}}_t||_{\bar{\mathbf{P}}_t^{-1}}^2 + ||\mathbf{A}_t^\top\breve{\mathbf{m}}_t||^2 - \log|\mathbf{I} + \mathbf{A}_t^\top\bar{\mathbf{P}}_t\mathbf{A}_t| \right) \tag{103}$$

**conditional linear Gaussian log partition**  For a Gaussian distribution, the log-partition function in terms of the natural parameters, $\boldsymbol{\lambda} = (\mathbf{h}, \text{vec}(\mathbf{J}))$, is given by

$$A(\boldsymbol{\lambda}) = \tfrac{1}{2}\mathbf{h}^\top\mathbf{J}^{-1}\mathbf{h} - \tfrac{1}{2}\log|\mathbf{J}| \tag{104}$$

so that, when

$$\mathbf{F} = \begin{pmatrix} \mathbf{Q}^{-1}\mathbf{A} & \mathbf{0} \\ \mathbf{0} & \mathbf{0} \end{pmatrix} \qquad \mathbf{f} = \begin{pmatrix} \mathbf{0} \\ \mathbf{Q}^{-1} \end{pmatrix} \tag{105}$$

and $\mathbf{u}^\top = \begin{pmatrix} \mathbf{u}_1^\top & \text{vec}(\mathbf{u}_2)^\top \end{pmatrix}$ is an arbitrary constant, we can write $A(\mathbf{F}t(\mathbf{z}) + \mathbf{f} + \mathbf{u}) = \mathbf{a}^\top t(\mathbf{z}) + b$ for some $\mathbf{a}$ and $b$. To show this, we can just expand the log partition function

$$A(\mathbf{F}t(\mathbf{z}) + \mathbf{f} + \mathbf{u}) = \tfrac{1}{2}\mathbf{z}^\top\mathbf{A}^\top\mathbf{Q}^{-1}(\mathbf{Q}^{-1} + \mathbf{u}_2)^{-1}\mathbf{Q}^{-1}\mathbf{A}\mathbf{z} + \mathbf{z}^\top\mathbf{A}^\top\mathbf{Q}^{-1}(\mathbf{Q}^{-1} + \mathbf{u}_2)^{-1}\mathbf{u}_1 \tag{106}$$

$$\quad + \tfrac{1}{2}\mathbf{u}_1^\top(\mathbf{Q}^{-1} + \mathbf{u}_2)^{-1}\mathbf{u}_1 - \tfrac{1}{2}\log|\mathbf{u}_2 + \mathbf{Q}^{-1}|$$

$$= \mathbf{a}^\top t(\mathbf{z}) + b \tag{107}$$

then $\mathbf{a}$ and $b$ can be identified as

$$\mathbf{a} = \begin{pmatrix} \mathbf{A}^\top\mathbf{Q}^{-1}(\mathbf{Q}^{-1} + \mathbf{U}_2)^{-1}\mathbf{u}_1 \\ -\mathbf{A}^\top\mathbf{Q}^{-1}(\mathbf{Q}^{-1} + \mathbf{U}_2)^{-1}\mathbf{Q}^{-1}\mathbf{A} \end{pmatrix} \quad b = \tfrac{1}{2}\mathbf{u}_1^\top(\mathbf{Q}^{-1} + \mathbf{u}_2)^{-1}\mathbf{u}_1 - \tfrac{1}{2}\log|\mathbf{Q}^{-1} + \mathbf{u}_2| \tag{108}$$

Then, for a LGSSM, we can evaluate the following expression which describes the difference between predictive and smoothed marginals,

$$\boldsymbol{\lambda}_t - \bar{\boldsymbol{\lambda}}_t = \boldsymbol{\alpha}_t + \nabla_{\boldsymbol{\mu}_t}\mathbb{E}_{q(\mathbf{z}_t)}\left[ A\left(\boldsymbol{\lambda}_{\boldsymbol{\theta}}(\mathbf{z}_t) + \boldsymbol{\lambda}_{t+1} - \bar{\boldsymbol{\lambda}}_{t+1}\right) - A\left(\boldsymbol{\lambda}_{\boldsymbol{\theta}}(\mathbf{z}_t)\right) \right] \tag{109}$$

$$= \boldsymbol{\alpha}_t + \begin{pmatrix} \mathbf{A}^\top\mathbf{Q}^{-1}(\mathbf{Q}^{-1} + \mathbf{P}_{t+1}^{-1} - \bar{\mathbf{P}}_{t+1}^{-1})^{-1}(\mathbf{h}_{t+1} - \bar{\mathbf{h}}_{t+1}) \\ \mathbf{A}^\top\mathbf{S}_{t+1}\mathbf{A} \end{pmatrix} \tag{110}$$

$$= \boldsymbol{\alpha}_t + \boldsymbol{\beta}_{t+1} \tag{111}$$

where $\mathbf{S}_{t+1} = \mathbf{Q}^{-1} - \mathbf{Q}^{-1}(\mathbf{Q}^{-1} + \mathbf{P}_{t+1}^{-1} - \bar{\mathbf{P}}_{t+1}^{-1})^{-1}\mathbf{Q}^{-1}$.

## G Forward KL fixed point

Using the moment matching property of the fixed point solution to the forward KL objective, we can write

$$\bar{\boldsymbol{\mu}}_{t+1}^* = \int t(\mathbf{z}_{t+1}) \, \mathbb{E}_{q_t} \left[ p_{\boldsymbol{\theta}}(\mathbf{z}_{t+1} \,|\, \mathbf{z}_t) \right] \mathrm{d}\mathbf{z}_{t+1} \tag{112}$$

$$= \int t(\mathbf{z}_{t+1}) \tag{113}$$

$$\times \left[ \int h(\mathbf{z}_t) h(\mathbf{z}_{t+1}) \exp \left( t(\mathbf{z}_{t+1})^\top \boldsymbol{\lambda}_{\boldsymbol{\theta}}(\mathbf{z}_t) + t(\mathbf{z}_t)^\top \boldsymbol{\lambda}_t - A(\boldsymbol{\lambda}_t) - A(\boldsymbol{\lambda}_{\boldsymbol{\theta}}(\mathbf{z}_t)) \right) \mathrm{d}\mathbf{z}_t \right] \mathrm{d}\mathbf{z}_{t+1}$$

$$= \int h(\mathbf{z}_t) \exp \left( t(\mathbf{z}_t)^\top \boldsymbol{\lambda}_t - A(\boldsymbol{\lambda}_t) \right) \tag{114}$$

$$\times \left[ \int t(\mathbf{z}_{t+1}) h(\mathbf{z}_{t+1}) \exp \left( t(\mathbf{z}_{t+1})^\top \boldsymbol{\lambda}_{\boldsymbol{\theta}}(\mathbf{z}_t) - A(\boldsymbol{\lambda}_{\boldsymbol{\theta}}(\mathbf{z}_t)) \right) \mathrm{d}\mathbf{z}_{t+1} \right] \mathrm{d}\mathbf{z}_t$$

$$= \mathbb{E}_{q_t} \left[ \boldsymbol{\mu}_{\boldsymbol{\theta}}(\mathbf{z}_t) \right] \tag{115}$$

This result holds for any setting where stochastic transitions, $p_{t+1|t}$, and the approximate marginal distributions, $q_t$, are restricted to the same exponential family distribution.

## H Linear Gaussian (information form) smoothing

Using the derived relation,

$$\mathbf{P}_t^{-1} = \bar{\mathbf{P}}_t^{-1} + \mathbf{C}^\top \mathbf{R}^{-1} + \mathbf{F}^\top \mathbf{S}_{t+1} \mathbf{F} \tag{116}$$

means that,

$$\mathbf{P}_t^{-1} - \bar{\mathbf{P}}_t^{-1} = \mathbf{C}^\top \mathbf{R}^{-1} \mathbf{C} + \mathbf{F}^\top \mathbf{S}_{t+1} \mathbf{F} \tag{117}$$

so that using the definition of $\mathbf{S}_t$ (rewritten for convenience) then plugging in, we get,

$$\mathbf{S}_t = \mathbf{Q}^{-1} - \mathbf{Q}^{-1} (\mathbf{Q}^{-1} + \mathbf{P}_t^{-1} - \bar{\mathbf{P}}_t^{-1})^{-1} \mathbf{Q}^{-1} \tag{118}$$

$$= \mathbf{Q}^{-1} - \mathbf{Q}^{-1} (\mathbf{Q}^{-1} + \mathbf{C}^\top \mathbf{R}^{-1} \mathbf{C} + \mathbf{F}^\top \mathbf{S}_{t+1} \mathbf{F}) \mathbf{Q}^{-1} \tag{119}$$

which gives a recurrence backward in time for $\mathbf{S}_{1:T}$ starting from $\mathbf{S}_{T+1} = \mathbf{0}$. Now since,

$$\mathbf{h}_t - \bar{\mathbf{h}}_t = \mathbf{F}^\top \mathbf{Q}^{-1} (\mathbf{Q}^{-1} + \mathbf{P}_{t+1}^{-1} - \bar{\mathbf{P}}_{t+1}^{-1})^{-1} (\mathbf{h}_{t+1} - \bar{\mathbf{h}}_{t+1}) \tag{120}$$

and,

$$(\mathbf{Q}^{-1} + \mathbf{C}^\top \mathbf{R}^{-1} \mathbf{C} + \mathbf{F}^\top \mathbf{S}_{t+1} \mathbf{F})^{-1} = \mathbf{Q} - \mathbf{Q} \mathbf{S}_t \mathbf{Q} \tag{121}$$

we have that,

$$\mathbf{F}^\top \mathbf{s}_t = \mathbf{F}^\top \mathbf{Q}^{-1} (\mathbf{Q}^{-1} + \mathbf{C}^\top \mathbf{R}^{-1} \mathbf{C} + \mathbf{F}^\top \mathbf{S}_{t+1} \mathbf{F})^{-1} (\mathbf{C}^\top \mathbf{R}^{-1} \mathbf{y}_{t+1} + \mathbf{F}^\top \mathbf{s}_{t+1}) \tag{122}$$

which simplifies into a recursion for $\mathbf{s}_{1:T}$ backward in time starting from $\mathbf{s}_{T+1} = \mathbf{0}$, given by,

$$\mathbf{s}_t = \mathbf{Q}^{-1} (\mathbf{Q}^{-1} + \mathbf{C}^\top \mathbf{R}^{-1} \mathbf{C} + \mathbf{F}^\top \mathbf{S}_{t+1} \mathbf{F})^{-1} (\mathbf{C}^\top \mathbf{R}^{-1} \mathbf{y}_{t+1} + \mathbf{F}^\top \mathbf{s}_{t+1}) \tag{123}$$

$$= (\mathbf{I} - \mathbf{S}_t \mathbf{Q}) (\mathbf{C}^\top \mathbf{R}^{-1} \mathbf{y}_{t+1} + \mathbf{F}^\top \mathbf{s}_{t+1}) \tag{124}$$

