# OpenReview forum: "eXponential FAmily Dynamical Systems (XFADS): Large-scale nonlinear Gaussian state-space modeling"
_NeurIPS.cc/2024/Conference — NeurIPS 2024 poster_

### Official Review · Reviewer_ehDs · 2024-06-30

**Soundness:** 3
**Presentation:** 3
**Contribution:** 3
**Rating:** 6
**Confidence:** 4

**Summary:**

This work develops a deep state-space model (DSSM) for state inference in and system identification of non-linear dynamical systems.
The objective function for learning the DSSM parameters is based on a smoothing formulation and thus the evidence lower bound (ELBO). The prior and likelihood parts are separately parametrized and joined in the natural parameter space, where the posterior natural parameters are simply the sum of prior and likelihood natural parameters. Furthermore, a low-rank approximation of the approximate likelihood (via encoder) are used to improve computational efficiency. Experiments are performed on synthetic data (pendulum and bouncing ball) as well as neuroscientific data (motor cortex readings of monkeys). The authors compare mostly against pretty old baselines from before 2016 (DVBF, DKF, SVAE).

**Strengths:**

- The paper is very well written and quite easy to follow for the most part (see weaknesses).
- Hyperparameter, notation and algorithm details as well as derivations are given in the supplementary material, which is very helpful.

**Weaknesses:**

- The mentioned limitations of previous work's limitations are not convincing. Fast evaluation of the loss function is hardly the problem that previous DSSM approaches tried to tackle. The main challenges of the VAE-based approaches is that they often only learn to auto-encode and do not properly identify the correct dynamics [1]. Indeed, tackling this problem seems to be a goal of the present work as well, but it is not argued in the introduction.
- The related work section also seemed to stop past 2016. There are multiple subsequent works that address and improve upon the problem of learning dynamics from high-dimensional data, see e.g. [2-5], all of which share several ideas with this work.

Soundness:
I have difficulties with the technical soundness of the paper, especially about the following:
- Eq. (6) is slightly incorrect: The expectation is taken only over z_{t-1}, however, the terms inside the expectation involve z_{t-1} and z_{t}. A correct smoothing-based objective involves pairs two-slice marginals, see e.g. [6]. Unfortunately, the paper is based on this equation and it does not seem to be just a typo as subsequent equation (e.g. (19)) have the same error.
- What exactly is the pseudo-observation \tilde{y}? What values does it take compared to an actual y? Previous work [2,4,5] use the term pseudo-observation to refer to an additional latent variable that encodes information about a single time-step obervation y_t. This makes it a bit confusing. As far as I understood, the \tilde{y} is not ever actually instantiated and instead used as a proxy for a likelihood approximation.
- line 138: I can not follow this argument. Why is smoothing turned into a filtering problem? Filtering and smoothing solve different problems. Do you instead mean that you have a backward-forward smoothing approach? If so, this needs to be explained and defined better. I am having trouble with the equations (14), (15) and several subsequent equations involving \tilde{y}, again mostly because I do not know what exactly these pseudo-observations are.
- It becomes more confusing that the introduction of a different distribution pi(z_t) is supposed to help. Please, can you elaborate on this part? How does the introduction of this distribution in the filtering formulation yield a smoothing/ELBO-based objective?

Novelty:
- It is not clear how this work compares to some of the more recent works that try to resolve issues with works prior to 2016. I referenced below already some closely related works that I am aware of, but there are several others missing in related work. It is not clear how this work is positioned in comparison to those.
- eq (11): using the sum of the two terms (prior and likelihood) is not novel, see e.g. [4], section 3.3, where a Gaussian likelihood approximation is combined with the Gaussian prior dynamics, leading to a sum of corresponding natural parameters.

Experiments:
- The experimental evaluation is rather weak. There are no proper benchmarks against more recent works. The qualtative results visualized in Figure 3 do not look that great and it is difficult to estimate how hard it is to predict this kind of data.


[1] Karl et al. 2016, Deep Variational Bayes Filters: Unsupervised Learning of State Space Models from Raw Data.
[2] Fraccaro et al. 2017, A Disentangled Recognition and Nonlinear Dynamics Model for Unsupervised Learning.
[3] Becker-Ehmck et al. 2019, Switching Linear Dynamics for Variational Bayes Filtering.
[4] Kurle et al. 2020, Deep Rao-Blackwellised Particle Filters for Time Series Forecasting.
[5] Klushyn et al. 2021, Latent Matters: Learning Deep State-Space Models.
[6] Simo Sarkka, Bayesian filtering and smoothing.

-------
Minors and suggestions that did not influence the review score
- The complexity statement in the introduction does not explain what the variables T, L, S, and R are. In my opinion, it would be better to describe it in words in the introduction with an emphasis on how it compares to other approaches.
- line 45: filtering is an even more common approximation than smoothing. Filtering, smoothing and even fixed-lag smoothing are all valid applications, corresponding to whether inference needs to be online, offline or semi-online.
- line 48: vEM needs a citation. For instance, VEM is mentioned in the dissertation of Beal, but there is likely a better reference therein.
- line 51: to make the difference to the VAE approach more clear, you might consider emphasizing that the alternative is a *coordinate ascent* approach with *alternating* optimization steps.
- line 60:  which favorable properties are the ones relevant to this work? I suppose the additive property or anything else? It should be made clear what exactly you refer to.
- line 68) typo: ii) -> iii)
- Equation (3) typo: parameter
- line 78: citation style changes
- dKF, dVBF, vEM all written with lower case first letter, but other works (including the original ones) use capitalization.

**Questions:**

- what is the difference between predictions from filtering and smoothing in Fig. 3? The last filter state should be identical to the last smoothing state.

**Limitations:**

Limitations have been addressed to some extent.

---

> ### Author Rebuttal · Authors · 2024-08-06
>
> Thank you for taking the time to review our manuscript.  Below we respond to your suggestions/questions starting with the weaknesses section.
>
> > The mentioned limitations of previous work's limitations are not convincing. Fast evaluation of the loss function is hardly the problem that previous DSSM approaches tried to tackle.
>
> Thank you for your remark — one of the reasons fast evaluation of the loss and approximate posterior statistics is not a problem for previous DSSM approaches is that they restrict the approximate posterior covariance to be diagonal.  When the diagonal restriction is dropped, then evaluating the KL terms requires linear algebraic operations (log-det, vector multiplication with matrix inverse, trace) that scale cubic with latent dimensionality, $L$.  However, structure in the posterior covariance because of our particular sample approximation and low-rank updates make it possible to carry out those operations in linear time.
>
> > The related work section also seemed to stop past 2016
>
> Thank you for pointing this out – we are aware of those works and there are many others as well we would have wished to include in the related works section (see [1a,2a,3a] below to name a few) but were ultimately limited by space.
>
> We highlighted particular works we did because we aim at general purpose approach with minimum assumptions about the dynamics equipped a scalable inference algorithm; we're not focusing on arguing for any apriori latent structures or dynamics model that are tailored for particular problems. We demonstrated with simple yet general networks to showcase that with a good inference method you can still achieve sota results.  Motivated by your comment, we will slightly condense the current related works section so that we can briefly touch upon recent works such as [1-5 and 1a-3a]
>
> > Eq. (6) is slightly incorrect: The expectation is taken only over z_{t-1}, however, the terms inside the expectation involve z_{t-1} and z_{t}.
>
> We apologize that space was tight and we couldn’t add a few more lines in the main text, but the expectations for Eq(6) do contain the two slice marginals.  Factorizing the variational posterior as $q(z_{1:T}) = q(z_1) \prod q(z_t \mid z_{t-1})$, we can write the standard ELBO as follows to arrive at Eq(6) involving the expected KL.
>
> $$  \mathcal{L}(q) = \int q(z_{1:T}) \log \frac{p(y_{1:T}, z_{1:T})}{q(z_{1:T})}  \, dz_{1:T}$$
> $$= \sum \int q(z_t) \log p(y_t\mid z_t) \, dz_t - \sum \int q(z_{1:T}) \log \frac{q(z_t\mid z_{t-1})}{p(z_t\mid z_{t-1})}  \, dz_{1:T}$$
> $$= \sum \int q(z_t) \log p(y_t\mid z_t) \, dz_t - \sum \int q(z_{t-1}) q(z_t\mid z_{t-1}) \log \frac{q(z_t\mid z_{t-1})}{p(z_t\mid z_{t-1})}  \, dz_{t-1, t}$$
> $$= \sum \mathbb{E}\_{q_t}   \left[ \log p(y_t \mid z_t) \right] -  \sum \mathbb{E}\_{q_{t-1}}  \left[ \mathbb{D}\_{\text{KL}}(q(z_t\mid z_{t-1}) || p(z_t \mid z_{t-1}) )\right]] $$
>
> For the same smoothing objective in an alternative paper see [4a] Eq.(6).
>
> > What exactly is the pseudo-observation \tilde{y}?
>
> The pseudo observation $\tilde{y}\_t$ encodes current/future observations, $y_{t:T}$, into a Gaussian potential.  Since it is a Gaussian potential, it is specified by parameters that interact linearly with the sufficient statistics – the form of those parameterizations are given explicitly by Eq (25) in the main text.  In the text, we tried to make this distinction by writing “Importantly, pseudo-observations deﬁned this way encode the current and future observations of the raw data – an essential component for transforming the statistical smoothing problem into an alternative ﬁltering problem."
>
> > line 138: I can not follow this argument. Why is smoothing turned into a filtering problem?  It becomes more confusing that the introduction of a different distribution pi(z_t) is supposed to help. Please, can you elaborate
>
> Thank you for the question – we say smoothing is turned into a filtering problem because pseudo observations, $\tilde{y}\_t$, encode $y_{t:T}$ into a single Gaussian potential, meaning we can filter pseudo observations to obtain statistics of the smoothed posterior; in other words, we have that $\pi(z_t) \approx  p(z_t \mid \tilde{y}\_{1:t}) \approx p(z_t \mid y_{1:T})$ .  The reason that we introduce $\pi(z_t)$ is because we want tractable Gaussian marginal approximations, and $q(z_t \mid z_{t-1})$ is conditionally Gaussian.
>
> > eq (11): using the sum of (prior and likelihood) is not novel
>
> Thank you for pointing this out, but we respectfully disagree that this particular parameterization is not novel.  We were inspired by works like [5a] and also [4], to use this type of parameterization; but in those works the dynamics are conditionally linear and it is straightforward to apply Gaussian message passing and recover the posterior – here the dynamics we consider are arbitrary and nonlinear, which led us to develop the nonlinear filtering procedure given by Eqs.(27) & (28) to make this possible.
>
> > what is the difference between predictions from filtering and smoothing in Fig. 3?
>
> You are correct the last filtering and smoothing state should be identical.  Our purpose was to show what decoded hand position/latents would look like during filtering in the case of a streaming data setting.
>
> [1a] Ansari et al, 2023. Neural Continuous-Discrete State Space Models for Irregularly-Sampled Time Series.
> [2a] Li et al, 2021. Scalable gradients for stochastic differential equations.
> [3a] Schimmel et al, 2022. iLQR-VAE : control-based learning of input-driven dynamics with applications to neural data.
> [4a] Krishnan et al 2016, Structured Inference Networks for Nonlinear State Space Models
> [5a] Johnson et al 2016, Composing graphical models with neural networks...
>
> Thank you again for your thoughtful suggestions and useful comments.  We hope if we have addressed them you might raise your score to reflect that.  We are very motivated to position our paper as best possible, and appreciate all of your helpful input.

---

> > ### Comment · Reviewer_ehDs · 2024-08-11
> > **Response to rebuttal**
> >
> > Thank you for your reply.
> >
> > > one of the reasons fast evaluation of the loss and approximate posterior statistics is not a problem for previous DSSM approaches is that they restrict the approximate posterior covariance to be diagonal.
> >
> > I agree with this and the favorable scaling is indeed a great contribution.
> > However, I still think that the introduction does not do a great job in introducing the problems of this research area, how this work fits in this area and how it addresses short-comings of the previous works.
> > What is missing is how previous works such as DVBF and DKF are insufficient (due to the diagonal approximation), that subsequent work such as [2,4,5] addresses this issue via conditional linearity and closed-form Kalman filtering/smoothing, how that subsequent work is insufficient, and how this work addresses this limitation.
> >
> > > [...] the expectations for Eq(6) do contain the two slice marginals. [...]
> >
> > You are right, the second expectation I thought was missing is of course contained in the KL. So I agree that Eq. (6) is indeed correct.
> >
> > > The pseudo observation encodes current/future observations into a Gaussian potential.
> >
> > Then I do not understand Eq. (12). Surely, you can approximate the likelihood of the actual data into a Gaussian potential. This is also what is used in the two-filter smoothing formula for linear dynamical systems: you collect the current and future data into a Gaussian potential in the form of the backwards-message, typically denoted by \beta = p(z_t | y_{t:T}). It is also possible to define the smoothing distribution via a backward-forward recursion, using that same \beta. The \beta message will be represented using the natural parameters of this Gaussian potential. Is this what you are doing here?
> > What I don't understand is why you need any pseudo-targets for this and whether these pseudo-targets that take concrete values actually exist. Why not just write \beta or something like g(z_t; y_{t:T}). Or am I missing something? Are these pseudo-targets used for anything? To underscore my point, note also that the RHS of Eq. (12) does not contain any \tilde{y}.
> >
> > > we say smoothing is turned into a filtering problem because pseudo observations \tilde{y}_t, encode y_{1:t} into a single Gaussian potential, meaning we can filter pseudo observations to obtain statistics of the smoothed posterior;
> >
> > As mentioned above, this looks to me like you are doing a backward-forward algorithm for smoothing. But you are right then with the formulation that in this case smoothing is turned into a filtering problem, however I still find pseudo-targets confusing here and I think it is easier to work with the Gaussian potentials or backward messages directly.
> >
> > > Thank you for pointing this out, but we respectfully disagree that this particular parameterization is not novel. [...] here the dynamics we consider are arbitrary and nonlinear, which led us to develop the nonlinear filtering procedure given by Eqs.(27) & (28) to make this possible.
> >
> > Agreed.

---

> > > ### Author Response · Authors · 2024-08-12
> > >
> > > We thank the reviewer for their comments and respond below.
> > >
> > > > However, I still think that the introduction does not do a great job in introducing the problems of this research area, how this work fits in this area and how it addresses short-comings of the previous works.  What is missing is how previous works such as DVBF and DKF are insufficient (due to the diagonal approximation), that subsequent work such as [2,4,5] addresses this issue via conditional linearity and closed-form Kalman filtering/smoothing, how that subsequent work is insufficient, and how this work addresses this limitation.
> > >
> > > We generally agree with the reviewer that some of the relevant advances need to be contrasted to form a better bigger picture for the reader. There are indeed many methods we have not included due to space. The switching LDS style systems such as [4] (or similarly rSLDS and its extensions) are good approximate representations of nonlinear state-space models, though we argue that it is not the naturally interpretable one when continuous state space is assumed (as in our neuroscience experiments). Since [4] is a particle filter like solution, it potentially has a tighter bound like [Zhao et al. 2022] which is another variational filter with constant time complexity. While [5] shares some goals of our work, similar to [4] it faces scalability issue in L.  As the reviewer points out, our contribution on the scalability front is solid; for example in  [5], the experiments were very small with L=3, 4, 5, and T = 15, 30, while we have L = 40 and T = 35, 130. Moreover, in contrast to previous works, we demonstrate the efficacy of our approach for causal filtering, which is immensely important for conducting causal investigations in neuroscientific settings. To the best of our knowledge, this is the only approach that can do so in large L scenarios.  Unfortunately neither [4] nor [5] have publicly available code – we will contact the authors so that we can make proper comparisons; our code is already publicly available, and if accepted, the camera ready will have a link to our codebase to support future investigations and reproducibility.
> > >
> > > We would like to re-emphasize that our variational inference network structure and the associated novel ELBO (Eq.22) are non-trivial contributions that work together to enable the scalability, predictive performance, principled masking, and the causal real-time filtering.  While we agree that SVAE and [2] both can have non-diagonal covariance, they are both based on a linear dynamical system (LDS) at the core – limiting their expressive power substantially; neuroscience data similar to those examined in the manuscript often exhibit dynamics with topological features LDS cannot capture (e.g. multiple fixed points).
> > >
> > > We will ensure that the related works section of the revised manuscript clearly articulates how these additional works fit into the bigger picture and also disambiguate their differences as you suggest.
> > >
> > > > Then I do not understand Eq. (12). Surely, you can approximate the likelihood of the actual data into a Gaussian potential. This is also what is used in the two-filter smoothing formula for linear dynamical systems: you collect the current and future data into a Gaussian potential in the form of the backwards-message, typically denoted by \beta = p(z_t | y_{t:T}). It is also possible to define the smoothing distribution via a backward-forward recursion, using that same \beta. The \beta message will be represented using the natural parameters of this Gaussian potential. Is this what you are doing here (yes)? What I don't understand is why you need any pseudo-targets for this and whether these pseudo-targets that take concrete values actually exist. Why not just write \beta or something like g(z_t; y_{t:T}). Or am I missing something? Are these pseudo-targets used for anything? To underscore my point, note also that the RHS of Eq. (12) does not contain any \tilde{y}.
> > >
> > > We believe we are on the same page and apologize for the confusion. The pseudo-observation $\tilde{y}\_t$ is indeed not instantiated, and is a representation for the natural parameters of the Gaussian potential given by $p(z_t | y_{t:T})$ which we call $\tilde{\lambda}\_\phi(y_{t:T})$. The $\beta\_{t+1}$ and $\alpha\_t$, parameterizing natural parameter updates as in Eq(24), when additively combined form this backward message as described in Eq (23). We realize this notational overhead might cause confusion and if accepted, we will make sure to reduce the notational overhead in the camera ready version. We hope this clears up any confusion.
> > >
> > > Thank you again for your suggestions and comments; we greatly appreciate the time you have taken to help improve our manuscript.

---

> > > > ### Author Response · Authors · 2024-08-14
> > > > **[Zhao et al. 2022]**
> > > >
> > > > We forgot to include the full citation of the relevant paper we mentioned in the previous response:
> > > >
> > > > - Zhao, Y., Nassar, J., Jordan, I., Bugallo, M., & Park, I. M. (2022). Streaming Variational Monte Carlo. IEEE Transactions on Pattern Analysis and Machine Intelligence, 45(1), 1150–1161. https://doi.org/10.1109/TPAMI.2022.3153225

---

### Official Review · Reviewer_UuWx · 2024-07-09

**Soundness:** 3
**Presentation:** 3
**Contribution:** 3
**Rating:** 7
**Confidence:** 4

**Summary:**

Update post author rebuttal.
Thanks for the clarifications in the common and personal replies. I will raise the score to Accept, trusting that you will make the improvements you mention.
________

The paper presents a class of non linear state-space models whose dynamics are defined as exponential family distributions.
The variational approximation leverages distributions in the same exponential family as the prior, and its parameterization is defined as the sum of prior parameters with a learnable data dependent term.
Efficient filtering/smoothing is obtained with an approximate message passing algorithm that exploits the low rank structure of the variational distribution.
The model is tested on a number of simulated and real world applications and shows good predictive performances.

**Strengths:**

* The paper introduces a theoretically principled model that is quite flexible and can be used in a wide range of applications. As such I believe it can have impact in the neurips community (as long as the code is released)

* The model combines ideas from previous work, but is overall novel to the best of my knowledge

* The model can handle well missing data

* The choice of inference network allows scalable inference

* The theoretical section is dense but well explained (unlike the experimental one as noted below)

* The predictive performances are better than similar SOTA models

**Weaknesses:**

**Main comment**

I found the experiment section to be too rushed and therefore hard to understand. The authors should improve its clarity (in case you need space to fit in the 9 pages, you can move some of the the theoretical results in section 4 to the appendix).
There are several issues that taken together make the experimental quite hard to follow:

1. In figures 1 and 2 your method is presented as "ours (n)" and "ours (c)", while figure 3 uses "ours (a)" and "ours (c)" . These a, c and n versions are however not defined anywhere, so I'm not sure what I am looking at.

2. Overall the way the figures are combined is messy. Not all of them are mentioned and described in the main text, where they appear in the following quite random order: 2b, 2a, 1, 3c, 3b, 2d. Especially considering the fact that you leave a lot of the explanation to the caption of the figures, it's very hard for the reader to understand which parts of the figures are relevant to look at while reading the different parts of the experimental section. I suggest to rethink the structure of the figures, possibly making a single figure per experiment and making sure they are sufficiently discussed in the main text.

3. Figure 2c mentions the "DMFC RGS" dataset which is not defined in section 5, so not sure what experiment that refers to. Figure 2c is also never mentioned in the main text.

4. In the caption of figure 2a "(top) for fixed S and (bottom) for fixed r" seems wrong.


**Other minor points**

* Line 36 in the introduction: the complexity term used terms that are not yet defined

* The method name (XFADS) only appears in the title and the discussion, but it is never introduced in the Method section, or used in the experimental section. Either you use it, or you can avoid introducing it

* Line 145 - you reference the non existing equation (94)

* Line 330, missing "the" in "if the number of samples"

* Is the format of the citations as a superscript an acceptable one for neurips? It's not very common

**Questions:**

Based on what I wrote in weaknesses section, I would like some clarification on my comments and know how the authors plan to improve the clarity of the experimental section

**Limitations:**

Yes

---

> ### Author Rebuttal · Authors · 2024-08-06
>
> We thank the reviewer for their helpful suggestions.  We apologize about the density of the paper at times and appreciate that you mentioned the possibility of moving some algorithmic discussion to the appendix in exchange for higher level intuitions and more in depth discussion pertaining to the experimental results.  We wholly agree, and as we echoed in the global rebuttal statement we are taking those suggestions to heart and moving some details from the sampling
>
> > Based on what I wrote in weaknesses section, I would like some clarification on my comments and know how the authors plan to improve the clarity of the experimental section
>
> In addition to what we wrote previously addressing this, motivated by your suggestions, we are restructuring the manuscript to enhance clarity of the experimental section by:
>
> - Displaying figures in the same order which the experiments are presented in the text
> - Explaining the data in more detail, such as the neural data we used to demonstrate model efficacy in monkey reaching and monkey timing (DMFC-RSG) tasks
> - Interweaving more explanation of the figures into the main text so that important details are not only contained in the captions.
> - Moving some algorithmic discussion to the appendix and, in turn, fleshing out the significance of the experimental results and giving more exposition.
>
>
> > Is the format of the citations as a superscript an acceptable one for neurips? It's not very common
>
> Thank you for asking – we did consult the guidelines beforehand.
>
>
> We have to again thank the reviewer for motivating us to move some algorithmic discussion to the appendix in exchange for space that can be spent on clarifying the practical implications demonstrated in the experimental results section.  We hope if we’ve adequately addressed your concerns you might raise your score to reflect that.  We are motivated to continue improving the manuscript, and appreciate all of the input you provided.

---

> > ### Comment · Reviewer_UuWx · 2024-08-10
> > **Score raised**
> >
> > Thanks for the clarifications in the common and personal replies. I will raise the score to Accept, trusting that you will make the improvements you mention.

---

> > > ### Author Response · Authors · 2024-08-11
> > >
> > > Thank you very much for your thoughtful consideration and for raising the score to Accept. We are grateful for your positive feedback and are committed to making the improvements you mentioned. Your insights have been incredibly valuable in refining our work, and we will ensure that the final version addresses all points raised in the review.

---

### Official Review · Reviewer_S7ai · 2024-07-12

**Soundness:** 4
**Presentation:** 4
**Contribution:** 3
**Rating:** 6
**Confidence:** 3

**Summary:**

This paper introduces a method for scalable nonlinear Gaussian state-space modeling that relies on variational autoencoders and a low-rank covariance matrix assumption for efficient inference by optimizing an approximate variational lower bound. The authors describe the computational benefits of their method and inference scheme and showcase its effectiveness on multiple real-world applications, drawing connections to neuroscience.

**Strengths:**

The paper reads clearly and is well-organized. The authors produce convincing experiments, and a novel inference method that scales linearly with the dimensionality of the state space. The authors approximate the filtering distribution with a differentiable approximation for gradient-based updates. The appendix is concise, with relevant and clear background information. The authors clearly motivate their work using real-world applications and a thorough literature review.

**Weaknesses:**

No major complaints. The paper is dense and notation heavy at times. It is unclear to me why the authors make a connection to causal amortized inference, and whether this is a natural connection for the scope of the paper.

**Questions:**

How easily extendable is the method to non-gaussian state-space models? More discussion on general methodology?
What do the learned low-rank covariance matrices look like?
How does amortization affect the quality of the approximate posterior? What if T is small?

**Limitations:**

In lines 89-92, the authors list the limitations of amortized inference for state space models. Are there limitations more specific to this particular method that the authors omit?

---

> ### Author Rebuttal · Authors · 2024-08-06
>
> Thank you for taking the time to review the paper and your comments that will undoubtedly increase the quality of our manuscript.
>
> > The paper is dense and notation heavy at times.
>
> We apologize that the paper could be dense at times.  Motivated by this comment and others, in order to enhance the clarity of the manuscript's main take-aways, we will move non-essential algorithmic details to the appendix in favor of extending the discussions of the experimental results section.
>
> > It is unclear to me why the authors make a connection to causal amortized inference, and whether this is a natural connection for the scope of the paper.
>
>
> Thank you for this question about causal amortized inference. One of the design choices we made, making it possible to perform causal inference (which we feel has important possible implications in real data-analysis where XFADS can be applied), was segmenting local/backward encoders – in that way filtering can be performed causally using the local encoder; because other sequential VAE models do not make that distinction, we find the ability to perform causal inference a feature of our model that distinguishes it from other approaches.
>
> > How easily extendable is the method to non-gaussian state-space models?
>
> Thank you for the great question! The convenient part of working with general exponential family representations is that the inference algorithm we presented (sans the Gaussian specific parts) is agnostic to the choice of distribution so long as we have a way of evaluating $\lambda_{\theta}(z_{t-1})$ and $\mu_{\theta}(z_{t-1})$.
>
>
> > What do the learned low-rank covariance matrices look like?
>
> Thank you for the question, we did not include these in the main paper due to space constraints, but we feel it would be beneficial to have some example learned covariances from an experiment.  In the rebuttal PDF, we show an example trial from the bouncing ball experiment for the models that use nonlinear dynamics; its interesting to see how for example, contact with the wall results in spatially complex covariance structures in latent space that cannot be captured by the diagonal approximations.
>
>
> > How does amortization affect the quality of the approximate posterior? What if T is small?
>
> In the case of smaller datasets, smaller T,  or amortization networks that are not very expressive, the method could still suffer from overfitting/amortization gaps usually associated with VAEs.
>
> > In lines 89-92, the authors list the limitations of amortized inference for state space models. Are there limitations more specific to this particular method that the authors omit?
>
> Thank you for pointing this out.  We have added to the discussion section some comments on further limitations – “Furthermore, depending on generative model specifications, such as $L$, while the inference framework of Alg.1 is always applicable, modifications of the message passing procedure in Alg.2 might be necessary to maximize efficiency (e.g. if $L$ is small but $S$ is large)."  In addition to that, in the related works section we will mention that there will be overhead incurred using low-rank approximations as compared to diagonal ones, but significant savings compared to methods like SVAE that use exact Gaussian message passing.
>
> We appreciate the time and effort you put into your review.  We hope that if you feel we have addressed the issues raised you might raise your score to reflect that.

---

> > ### Comment · Reviewer_S7ai · 2024-08-11
> >
> > I thank the authors for their detailed, thorough rebuttal, and address some of their comments below.
> >
> > > Thank you for this question about causal amortized inference. One of the design choices we made, making it possible to perform causal inference (which we feel has important possible implications in real data-analysis where XFADS can be applied), was segmenting local/backward encoders – in that way filtering can be performed causally using the local encoder; because other sequential VAE models do not make that distinction, we find the ability to perform causal inference a feature of our model that distinguishes it from other approaches.
> >
> > Thank you for the clarification.
> >
> > > The convenient part of working with general exponential family representations is that the inference algorithm we presented (sans the Gaussian specific parts) is agnostic to the choice of distribution so long as we have a way of evaluating $\lambda_{\theta}(z_{t-1})$ and $\mu_{\theta}(z_{t-1})$.
> >
> > Thank you for the clarification. If I understand correctly, fast inference relies on the reparameterization trick for approximating the reconstruction term in the evidence lower bound. For exponential family distributions that are not reparametrizable, what variance reduction techniques do you recommend?
> >
> > > In the rebuttal PDF, we show an example trial from the bouncing ball experiment for the models that use nonlinear dynamics; its interesting to see how for example, contact with the wall results in spatially complex covariance structures in latent space that cannot be captured by the diagonal approximations.
> >
> > Very interesting. I request the authors include at least one example figure in the appendix (no need to compare to other methods here) so the readers may observe the low rank, non-diagonal covariance structures.
> >
> > > We have added to the discussion section some comments on further limitations – “Furthermore, depending on generative model specifications, such as $L$, while the inference framework of Alg.1 is always applicable, modifications of the message passing procedure in Alg.2 might be necessary to maximize efficiency (e.g. if $L$ is small but $S$ is large)." In addition to that, in the related works section we will mention that there will be overhead incurred using low-rank approximations as compared to diagonal ones, but significant savings compared to methods like SVAE that use exact Gaussian message passing.
> >
> > Thank you for the additional comments regarding the limitations of the work.

---

> > > ### Author Response · Authors · 2024-08-11
> > >
> > > We thank the reviewer for their prompt response and helpful suggestions.  We address your comments below.
> > >
> > > > Thank you for the clarification. If I understand correctly, fast inference relies on the reparameterization trick for approximating the reconstruction term in the evidence lower bound. For exponential family distributions that are not reparametrizable, what variance reduction techniques do you recommend?
> > >
> > > Thank you for the question.  You are right that fast inference relies on the reparameterization trick.  In cases where the standard reparameterization trick cannot be applied, an alternative (equivalent to extended kalman filter in the Gaussian case) would be to linearize the dynamics function in mean parameter space, $\mu\_{\theta}(z_{t-1})$, about the sufficient statistics $\mathcal{T}(z_{t-1})$ so that $\mu\_{\theta}(z_{t-1}) \approx F \mathcal{T}(z_{t-1}) + f$, letting us evaluate the predict step equation $\mathbb{E}[\mu\_{\theta}(z_{t-1})] \approx \mathbb{E}[F \mathcal{T}(z_{t-1}) + f] = F \mu\_{t-1} + f$.  Alternatively, it would also be viable to use implicit reparameterization gradients[1] when evaluating the ELBO.  Motivated by this question, we will include additional discussion related to these alternatives in the appendix.
> > >
> > > [1] Figurnov et al. 2018. Implicit reparameterization gradients
> > >
> > >
> > > > Very interesting. I request the authors include at least one example figure in the appendix (no need to compare to other methods here) so the readers may observe the low rank, non-diagonal covariance structures.
> > >
> > > Thank you for the original suggestion — we will make sure to include in the appendix of the manuscript the bouncing ball covariances from the rebuttal PDF as well as visualizations of the latent state covariance for some of the other experiments.
> > >
> > >
> > > Thank you again for your thoughtful questions and constructive feedback. We appreciate your time and effort in reviewing our response and believe the revisions will strengthen the manuscript.

---

### Official Review · Reviewer_e1Je · 2024-07-17

**Soundness:** 3
**Presentation:** 3
**Contribution:** 4
**Rating:** 7
**Confidence:** 3

**Summary:**

The paper presents a novel state space model (SSM) framework for learning dynamical systems with nonlinear latent dynamics. The proposed method borrows inspirations from structured variational autoencoders (SVAEs) and sample-based nonlinear Bayesian filtering, using low-rank corrections to the prior to capture information from the observations and compute approximations to the variational posterior. This framework addresses limitations in current methods which fall short on model expressivity or predictive power.

**Strengths:**

This paper builds upon previous work very well while addressing their weaknesses adequately. The authors show a high level of technical understanding regarding Bayesian filtering and variational inference, and from what I have read the methods proposed are technically sound (I did not go over all of the math in the appendix). As far as I’m aware, the proposal of a structured inference framework for exponential family SSMs that allows for learned non-linear dynamics is a good contribution that fills a gap in the literature. The paper is also professionally written and contains few mistakes or typos. Overall a good paper with solid contributions.

**Weaknesses:**

- Main technical ideas are conveyed in a convoluted way. The main paper is unnecessarily dense, making it hard to read and understand. One suggestion is to keep only the core equations (such as the objective (21)(22), the low rank parameterization (24), etc.) and provide more high-level discussions on the intuition behind the model. For example, the author could talk about the relationship between the pseudo-observations and the potentials in the SVAE paper, and give intuition on how to understand these quantities. Technical details such as the sample-based approximate inference and the corresponding low-rank updates are pretty standard in my opinion and should be left in the appendix.
- Some related work is missing. From my understanding, the authors use sampling-based linear Gaussian approximations for inference in the nonlinear dynamical system. This is closely related to the extended Kalman filter (EKF) which uses linear approximations to the dynamics based on the Jacobian, and the unscented Kalman filter (UKF) which uses anchor points instead of random samples to estimate the mean and covariance of the predicative distribution. There have also been attempts to combine neural networks with these nonlinear Bayes filtering methods such as (Liu et al., 2024).
- Somewhat limited experimental verification. While I do think that the existing results conveys the effectiveness of the proposed methods well, I do wish that more in-depth comparisons and discussions can be made. For example, how well do the low-rank update scale to higher-dimensional systems? Comparisons like this gives the reader a better sense of the tradeoffs of the method compared to others in the literature.

Overall, I consider these points to be relatively minor and do not detract from the paper's contributions.

Liu, Wei, et al. "Neural extended Kalman filters for learning and predicting dynamics of structural systems." *Structural Health Monitoring* 23.2 (2024): 1037-1052.

**Questions:**

- Can this method be efficiently parallelized?
- What are the practical computational efficiency of this method compared to the DKF or the SVAE in wall clock time? Are the computational overheads for the sampling-based inference significant in practice?
- In figure 1, the DVBF and SVAE seems to be underperforming the DKF in the pendulum experiments, despite the pendulum motion closely resembling a linear dynamical system. Why is this the case?
- How well do the low-rank update scale to higher-dimensional systems?

**Limitations:**

The authors have adequately addressed the limitations of their work.

---

> ### Author Rebuttal · Authors · 2024-08-06
>
> Thank you for taking the time and effort and asking questions that we feel have helped make the manuscript stronger.  Below we respond to them in order,
>
> >   One suggestion is to keep only the core equations (such as the objective (21)(22), the low rank parameterization (24), etc.) and provide more high-level discussions on the intuition behind the model. The author could talk about the relationship between the pseudo-observations and the potentials in the SVAE paper, and give intuition on how to understand these quantities.
>
> Thank you for this suggestion — we understand the paper is dense and building more intuition will benefit the reader. Motivated by your suggestion along with those of reviewer UuWx and others, we are moving some algorithmic details from section 3 to the Appendix in favor of higher level intuitions and clarity of the experimental results.
>
> >  Some related work is missing. From my understanding, the authors use sampling-based linear Gaussian approximations for inference in the nonlinear dynamical system. This is closely related to the extended Kalman filter (EKF) which uses linear approximations to the dynamics based on the Jacobian, and the unscented Kalman filter (UKF) which uses anchor points instead of random samples to estimate the mean and covariance of the predicative distribution.
>
> Thank you for bringing up other approximate filtering methods.  This led to the discovery that the approximate predict step of Eq(23), when dynamics are nonlinear Gaussian, coincides with the statistically linearized Kalman filter [1]  Thanks to your suggestion, we add an additional line after introducing the approximate filter to clarify this connection: “...otherwise, Monte Carlo integration can provide a differentiable approximation; for nonlinear and Gaussian dynamics this leads to a predict step equivalent to the statistically linearized Kalman filter [1].”
>
> > Somewhat limited experimental verification. While I do think that the existing results conveys the effectiveness of the proposed methods well, I do wish that more in-depth comparisons and discussions can be made. For example, how well do the low-rank update scale to higher-dimensional systems? Comparisons like this gives the reader a better sense of the tradeoffs of the method compared to others in the literature.
>
> These are great suggestions and as per the earlier remark, we are moving non-essential algorithmic details into the appendix.  We will be using the extra space to help build extra intuition – for example, elaborating on Fig.2(a,b) and emphasizing the favorable scaling illustrated in the figure.  We have also run experiments using the latent SDE of [3] as an additional point of comparison(Fig.1 rebuttal PDF).
>
> > What are the practical computational efficiency of this method compared to the DKF or the SVAE in wall clock time? Are the computational overheads for the sampling-based inference significant in practice?
>
> Thanks for this question -- it has motivated us to expand on this in the discussion section as well as add other methods wallclock time to Fig.1a for reference .  To answer: compared to methods like DKF/DVBF that use diagonal approximation there is additional overhead since our method scales $\mathcal{O}(LSr)$ per step, however, there are significant savings compared to SVAE which scales $\mathcal{O}(L^3)$ per step due to its dense marginal covariances.
>
> >  In figure 1, the DVBF and SVAE seems to be underperforming the DKF in the pendulum experiments, despite the pendulum motion closely resembling a linear dynamical system. Why is this the case?
>
> It is true that the pendulum closely resembles an LDS, but in this case, the small angle approximation does not hold [2] which would make long-term forecasting difficult for a model using a linear dynamical system prior.  This is one of the reasons that all methods perform very well in the smoothing metric but not prediction metric; especially SVAE where the linear dynamics approximation degrades as the horizon increases.
>
> > Can this method be efficiently parallelized?
>
> Things can be easily parallelized across batches, but to parallelize across time would require a different inference scheme (since we sequentially propagate samples through the dynamics).  However, one possibility is to use a parallelizable inference network architecture (such as S4 [4]), to produce marginals and train it as usual using Eq.(22)
>
> > How well do the low-rank update scale to higher-dimensional systems? I do wish that more in-depth comparisons and discussions can be made. For example, how well do the low-rank update scale to higher-dimensional systems? Comparisons like this gives the reader a better sense of the tradeoffs of the method compared to others in the literature
>
> Thank you for these questions – we aimed to elucidate some aspects through conducting the experiments featured in the left half of figure 2.   Motivated by your previous comment, we will now have more room to explain the significance of that experiment in showing: i) purposely exploiting covariance structures makes it possible to develop a filtering algorithm scaling linear in the latent dimension, L — compared to the O(L^3) cost per step incurred for typical Gaussian filters.   and ii) how low-rank covariance parameterizations can also lead to a tighter ELBO.
>
> We would like to thank the reviewer again for their time and useful suggestions that will serve to make our manuscript stronger.  We hope that if we have addressed your concerns you might raise your score to reflect that, and we are eager to address any concerns or questions that might remain.
>
> [1] Sarkka, Bayesian filtering and smoothing.
> [2] Zhao and Lidnerman, ICML 2023. Revisiting Structured Variational Autoencoders
> [3] Li et al, 2021. Scalable gradients for stochastic differential equations
> [4] Gu et al, 2021. Efficiently Modeling Long Sequences with Structured State Spaces

---

> > ### Comment · Area_Chair_96pP · 2024-08-13
> > **Last day for discussion**
> >
> > Reviewer e1Je, today is the last day for discussion. I hope you will respond to the authors' rebuttal.

---

> > ### Comment · Reviewer_e1Je · 2024-08-14
> >
> > I would like to thank the authors for their thorough response. I am glad that the authors find my suggestions on the paper organization and related works helpful. Trusting that the authors will make the promised changes and provide more discussion on the benefit and scaling properties of the low-rank approximation, I have raised my score to a 7.

---

### Author Rebuttal · Authors · 2024-08-06

First, we would like to thank all of the reviewers for their time, effort, and helpful comments regarding the submitted manuscript – we feel many of your suggestions have led us to changes and additions that better position the paper.

Many reviewers were positive about the clarity of writing, but felt that the paper was a bit dense at times and that room taken up by some algorithmic details could be used instead to deliver more insightful discussion in the experimental results section.  We agree with this sentiment and are very happy it was brought to our attention – after reading through all of the reviews, we will make the following changes that will help to elucidate the significance of the experimental results section.

- Some equations from the sample approximation structure will be delegated to the appendix which will help to i) create space for higher level insights and discussion and ii) keep focus of the reader away from these lower level details that might detract from the main message.
- We will use the extra space for extra discussion regarding the results;  for example, Fig.2a/b demonstrate the favorable scaling of our inference algorithm, how low-rank approximations scale, and convergence of the ELBO for different parameter settings – now we are able to drive these points home.
- As an additional point of comparison, we have run pendulum/bouncing-ball/monkey reaching experiments using the latent SDE method of [1] – for a total of 5 latent variable models that we compare against.


In the rebuttal figure, the additional results include:
- Collated results featuring the new results from applying the latent-SDE method to experiments considered in the paper
- Additional analysis of the model learned from the monkey reaching data set, showing the potential impact of having a highly predictive model when applied to motor control or brain computer interfaces.
- Examples of the learned posterior covariances of the nonlinear state-space methods in comparison with our own on the bouncing ball dataset.

We would also like to take some room to point out what we believe are important technical contributions, and the expected impact this work could have.

## technical contributions

To the best of our knowledge, the state-space variational autoencoding framework developed in this manuscript is the only one to allow the recovery of non-trivial covariance over space while scaling well to higher dimensions when general form (possibly nonlinear) dynamics are used.  Gaussian message passing naively scales O(L^3) per time step, but exploiting the sample approximation and low-rank update structures let us develop an O(LSr) complexity algorithm – as a further point, infinite horizon Kalman filters (which assume steady-state covariance) have been classically used when L becomes large and still scale O(L^2).  By being able to filter in O(LSr) time per step, we open the door for high-dimensional filtering with tunable fidelity knobs given by S (# of samples in predict step) and r (rank of update step precision update) without restrictions to trivial covariance structures.

We also view introducing the distinction of specialized local/backward natural parameter encoders in the context of state-space models as a novel contribution for several reasons.  i) This makes it possible to use the causal variant of the model after it has been trained, and deploy it in streaming data settings to perform online filtering with no modifications needed.  ii) As also discussed in the manuscript, this allows missing data to be seamlessly handled as a result of their natural parameter representation. iii) This distinction also makes it possible to avoid parameterizing a neural network for the local encoder when the observation model is conjugate to the dynamics, since the optimal local encoder could be found in closed form as $\alpha_t = \nabla_{\mu_t}\mathbb{E}\_{q_t}[\log p(y_t \mid z_t)]$.  We hope also this distinction can help lead to better architectural design choices for inference networks in the context of amortized inference.

## expected impact

Our algorithm and parameterization of the variational posterior allows for causal inference that is suitable for real-time applications and supports recovery of temporally causal states.  For example, in the manuscript we show how on numerous examples, the inference framework we propose learns models that are much more predictive than other deep state-space models. Applied to the monkey reaching data in Fig.3 we showed how when applied to real neural recordings our learned model was able to successfully *predict* reaching behavior (which it was not trained on).  We support these results further in the rebuttal PDF, showing accurate movement speed predicted from the model long before time of the movement onset.

Thank you again to all of the reviewers for their comments and carefully thought out suggestions.  We hope that we can have a productive dialogue and continue to address any remaining concerns that the reviewers might have and position this work as best as possible.


Best,
The authors

---

### Comment · Area_Chair_96pP · 2024-08-10
**Please respond to author comments**

Dear Reviewers,

Please read the authors’ rebuttal and respond to their comments. The discussion period ends on Aug 13 and I would like to have time for back and forth, as necessary.

---

### Decision · Program_Chairs · 2024-09-25

**Decision:**

Accept (poster)

**Comment:**

This manuscript presents a method for efficient, structure-exploiting, amortized variational inference in sequential variational autoencoders. The approach is based on a smoothing formulation of the variational posterior, along with some clever approximations that capitalize on the exponential family structure of the variational potentials. Moreover, the results on a variety of real and synthetic datasets look very impressive.

The math and notation are a bit heavy in places, and I encourage the authors to spend time editing for clarity. There are also a few places where large jumps are taken from one equation to the next, and I think the paper would draw in a broader audience if the main text was more approachable. Overall, though, I think this is a very nice contribution to the field.